# Neuronal selectivity to complex vocalization features emerges in the superficial layers of primary auditory cortex

Pilar Montes-Lourido[1,☉,¤], Manaswini Kar[1,2,☉], Stephen V. David[3], Srivatsun Sadagopan[1,2,4,5,*]

1 Department of Neurobiology, University of Pittsburgh, Pittsburgh, Pennsylvania, United States of America, 2 Center for Neuroscience, University of Pittsburgh, Pittsburgh, Pennsylvania, United States of America, 3 Department of Otolaryngology, Oregon Health and Science University, Portland, Oregon, United States of America, 4 Department of Bioengineering, University of Pittsburgh, Pittsburgh, Pennsylvania, United States of America, 5 Center for the Neural Basis of Cognition, University of Pittsburgh, Pittsburgh, Pennsylvania, United States of America

☉ These authors contributed equally to this work.
¤ Current address: Department of Transfer and Innovation, USC University Hospital Complex (CHUS), University of Santiago de Compostela, Spain
* vatsun@pitt.edu

**Data Availability Statement:** All relevant data are within the paper and its Supporting Information files.

## Abstract

Early in auditory processing, neural responses faithfully reflect acoustic input. At higher stages of auditory processing, however, neurons become selective for particular call types, eventually leading to specialized regions of cortex that preferentially process calls at the highest auditory processing stages. We previously proposed that an intermediate step in how nonselective responses are transformed into call-selective responses is the detection of informative call features. But how neural selectivity for informative call features emerges from nonselective inputs, whether feature selectivity gradually emerges over the processing hierarchy, and how stimulus information is represented in nonselective and feature-selective populations remain open question. In this study, using unanesthetized guinea pigs (GPs), a highly vocal and social rodent, as an animal model, we characterized the neural representation of calls in 3 auditory processing stages—the thalamus (ventral medial geniculate body (vMGB)), and thalamorecipient (L4) and superficial layers (L2/3) of primary auditory cortex (A1). We found that neurons in vMGB and A1 L4 did not exhibit call-selective responses and responded throughout the call durations. However, A1 L2/3 neurons showed high call selectivity with about a third of neurons responding to only 1 or 2 call types. These A1 L2/3 neurons only responded to restricted portions of calls suggesting that they were highly selective for call features. Receptive fields of these A1 L2/3 neurons showed complex spectrotemporal structures that could underlie their high call feature selectivity. Information theoretic analysis revealed that in A1 L4, stimulus information was distributed over the population and was spread out over the call durations. In contrast, in A1 L2/3, individual neurons showed brief bursts of high stimulus-specific information and conveyed high levels of information per spike. These data demonstrate that a transformation in the neural representation of calls occurs between A1 L4 and A1 L2/3, leading to the emergence of a feature-based

**Funding:** This work was supported by the National Institutes of Health, NIH R01DC017141, www.nih. gov, (SS); by the 2018 NARSAD Young Investigator grant, 27675, Brain and Behavior Research Foundation, https://www.bbrfoundation. org/, (SS) and by the Pennsylvania Lions Hearing Research Foundation, https://plhrf.org/ (SS). The funders had no role in study design, data collection and analysis, decision to publish, or preparation of the manuscript.

**Competing interests:** The authors have declared that no competing interests exist.

**Abbreviations:** AL, anterolateral; BOLD, blood–oxygen level–dependent; CSD, current source density; FDR, false discovery rate; FS, fast-spiking; GP, guinea pig; HSD, honestly significant difference; IC, inferior colliculus; K–S, Kolmogorov–Smirnov; LFP, local field potential; LN, linear–nonlinear; MI, mutual information; NEMS, Neural Encoding Model System; PSTH, peristimulus time histogram; RS, regular-spiking; STRF, spectrotemporal receptive field; USV, ultrasonic vocalization; vMGB, ventral medial geniculate body; VRB, ventral–rostral belt.

representation of calls in A1 L2/3. Our data thus suggest that observed cortical specializations for call processing emerge in A1 and set the stage for further mechanistic studies.

## Introduction

How behaviorally critical sounds, such as conspecific vocalizations (calls), are represented in the activity of neural populations at various stages of the auditory processing hierarchy is a central question in auditory neuroscience. Early representations of sounds, such as in the auditory nerve, have been proposed to be optimized for the efficient and faithful representation of sounds in general [1,2]. Consequently, at lower auditory processing stations, vocalizations are not represented any differently than other sounds ([3,4]; but see [5]). At the other extreme, behaviorally relevant stimuli such as vocalizations are overrepresented at the highest cortical processing stages [6–9]. In macaques and marmosets, neurons in the highest stages of the auditory processing hierarchy show strong selectivity for call category and even caller identity [10–12]. How the neural representation of calls is transformed from a nonspecific format in early processing stages to a call-selective format at higher processing stages remains unclear. Because auditory receptive fields increase in complexity as one ascends the auditory processing hierarchy [13,14], the conventional hypothesis is that call selectivity is gradually refined across auditory processing stages. However, there is little systematic evidence supporting a gradual refinement in call selectivity. While many studies have investigated call representations in subcortical and cortical stages [6,7,15–27], these have not systematically explored the mechanisms of how call representations could be transformed from one stage to the next or how this impacts information representation at different processing stages. A clear understanding of where critical transformations occur is an essential first step in designing experiments to probe neural mechanisms underlying these transformations and to target these experiments to the appropriate processing stage in the auditory hierarchy. In this study, we recorded neural responses to an extensive set of call stimuli across multiple auditory processing stages to test whether the emergence of call selectivity is gradual and to characterize the nature and informativeness of call representations at these processing stages.

The first question to address is what it means for a neuron to be call selective. In many mammalian species, calls are not produced stereotypically from trial to trial; rather, calls are instantiations of an underlying noisy production process. Thus, there is considerable variability in the production of calls belonging to a given call category both across trials and across individuals [28,29]. Furthermore, different call categories may have highly overlapping spectral content. To be call category selective, a neuron has to be selective for more than purely spectral cues and has to generalize across production variability. In previous theoretical work, we showed that in order to construct high level call category-selective neural responses, it is first necessary to have an intermediate representation where neurons detect informative call features [29]. Informative call features are spectrotemporal fragments of calls that are most likely to be found across exemplars of a given category (despite production variability) and typically span about an octave in frequency and about a hundred milliseconds in time. Thus, if one of the objectives of cortical processing is call categorization, our model would predict the existence of diverse neurons, each tuned for model-predicted informative features. Consistent with this prediction, limited experimental data suggested that call feature-selective neurons could be found in primary auditory cortex (A1) of marmosets and guinea pigs (GPs) [29]. But the question remains whether feature selectivity is gradually constructed over the ascending auditory pathway, or if it emerges de novo at some processing stage.

At lower processing stations of the auditory pathway in GPs and nonhuman primates, there is little evidence for the existence of call feature-selective neurons [15,16,22]. Rather, neurons appear to respond to call types in a manner largely explained by frequency tuning [15,16,22]. In GPs, single neurons in the inferior colliculus (IC) are not selective for particular call types or call features [16]. In primates and GPs, even at the level of A1, many previous studies have not reported strong selectivity for particular call types or features, or preference for natural over reversed calls ([17,20,21,30]; but see below). It is only at the level of secondary cortex that clear call-selective responses have been reported, both in primates (in anterolateral (AL) belt; [8,9]), and in GPs (Area S and the ventral–rostral belt (VRB) [6]). However, gaps in understanding remain because of some technical limitations of these studies, including the use of anesthesia, limited stimulus sets, multiunit recordings, or not comparing across processing stages, specifically across cortical laminae. Thus, these studies do not give rise to a clear picture of where and how a call feature-specific representation first emerges.

A few studies have provided hints that A1 could be a locus of important transformations to the neural representation of calls. In A1 of awake squirrel monkeys, one study reported that about a third of neurons responded to call stimuli that showed similarities in their frequency–time characteristics [23]. In marmoset A1, about a third of A1 neurons at shallower recording depths showed highly nonlinear receptive fields that could in turn underlie call feature selectivity [31]. It has been proposed that because A1 neurons cannot phase lock to fast envelope fluctuations, sparse spiking in A1 could provide temporal markers that reflect subcortical spectrotemporal integration [32]. But these studies did not specify whether recordings were from the input or output layers of A1. In humans, a recent study using ultrahigh field fMRI with laminar resolution reported that whereas blood–oxygen level–dependent (BOLD) activity in granular and infragranular layers could be explained using simple frequency content-based models, activity in supragranular layers could be explained better using more complex models incorporating spectral and temporal modulations [33]. This supragranular activity resembled activity in secondary auditory cortical areas, suggesting that a transformation between thalamorecipient (A1 L4) and superficial (A1 L2/3) layers of A1 might give rise to more specialized processing. Thus, a careful investigation of the thalamus and across identified cortical laminae of A1 is necessary to understand how the cortex might transform sound representations, particularly with respect to behaviorally critical sounds such as calls.

In this study, we begin to address how early nonspecific and spectral content-based representations are transformed into higher feature-based representations. We recorded neural activity from unanesthetized GPs passively listening to an extensive range of conspecific calls [6,34,35] and acquired single-unit responses from the thalamus (ventral medial geniculate body (vMGB)), thalamorecipient (A1 L4), and superficial (A1 L2/3) layers of A1. We found that neurons in vMGB and A1 L4 responded to most call categories and throughout the call durations. In contrast, a third of A1 L2/3 neurons responded sparsely and selectively to 1 or 2 call categories, and only in specific time bins within a call. These A1 L2/3 neurons showed highly complex receptive fields that could underlie this call feature selectivity. Information-theoretic analyses revealed that while average mutual information (MI) was high in A1 L4, MI was about evenly distributed over the population of neurons and across multiple stimuli and sustained over the stimulus duration. In contrast, individual A1 L2/3 neurons were highly informative about few stimuli and conveyed high levels of information per spike in only a handful of time bins. These results argue against a gradual emergence of call feature selectivity and suggest that a significant transformation in the neural representation of calls occurs between A1 L4 and A1 L2/3, leading to the emergence of a feature-based representation of calls in A1 L2/3.

## Results

We recorded the activity of single neurons located in the vMGB, A1 L4, and A1 L2/3 of unanesthetized, head-fixed, passively listening GPs (Fig 1A, top). We first implanted a head post and recording chambers onto the skull of the animals using aseptic surgical technique. We then performed small craniotomies (approximately 1.0 mm diameter) to access the underlying tissue (Fig 1A, bottom). Single-unit activity was recorded using high-impedance tungsten electrodes and first sorted online using a template match algorithm, and later refined offline. Over a few weeks, we sequentially recorded from a number of such craniotomies and

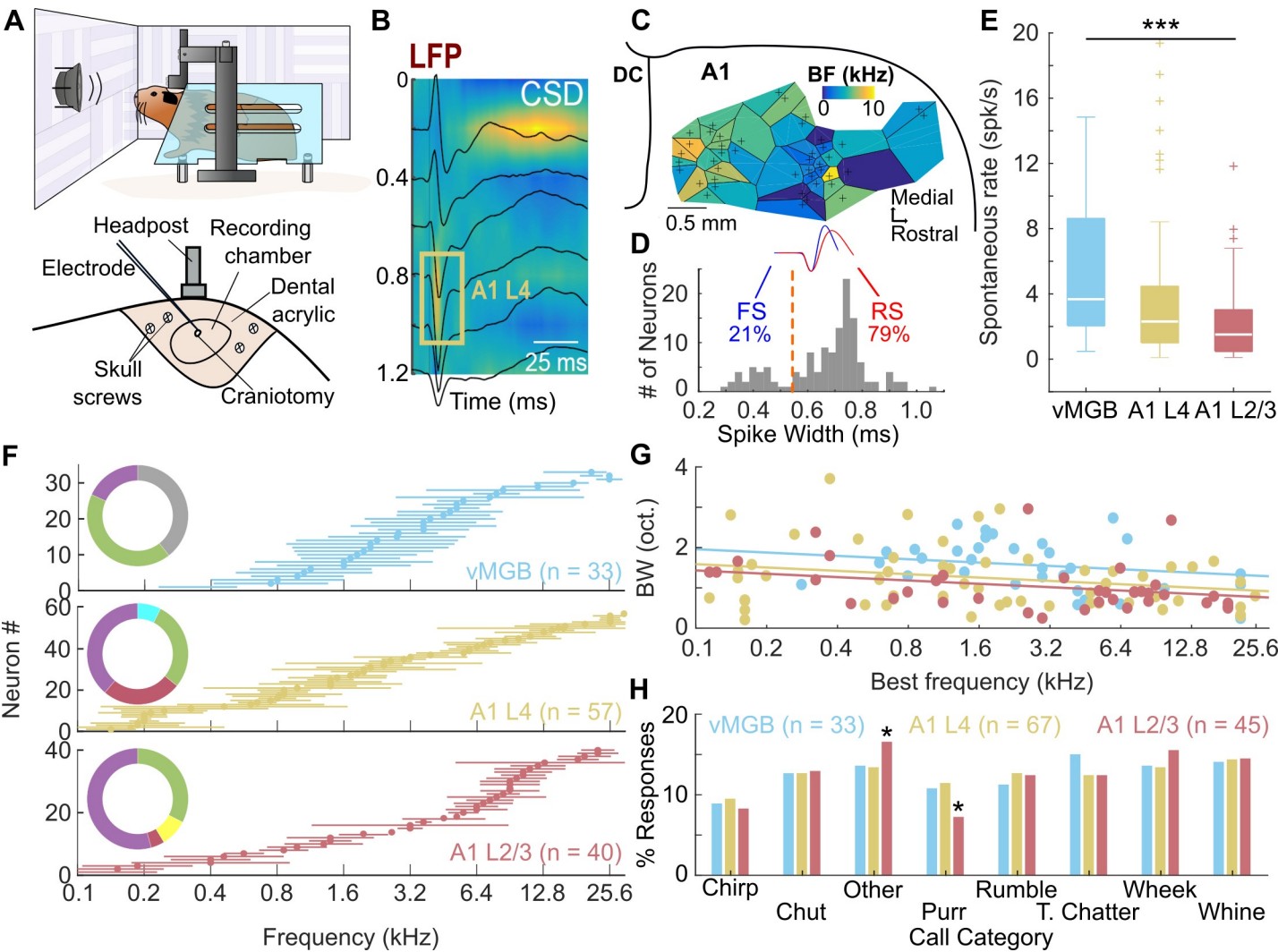

**Fig 1. Single-unit recordings from unanesthetized, head-fixed GPs.** (A) Recording setup (top) and details of cranial implant (bottom). (B) Average LFP traces (black lines) and CSD (colormap; warm colors correspond to sinks) of an example electrode track in A1. Yellow box outlines estimated A1 L4 location. (C) Example Voronoi map showing tonotopy of auditory cortex in one GP. Colormap corresponds to best frequency. (D) Histogram of spike widths of sorted single units. Dashed orange line is the threshold used to separate FS (blue) from RS (red) units. (E) Distribution of spontaneous rates in vMGB (blue), A1 L4 (yellow), and A1 L2/3 (red). ***$p < 0.005$, Kruskal–Wallis test (Dunn–Sidak post hoc test). (F) Best frequencies (discs) and bandwidths (lines) of tone-responsive neurons recorded from vMGB, A1 L4, and A1 L2/3 (colors as earlier). Insets show distribution of units across subjects, colors correspond to individual subjects. (G) Tone tuning bandwidth plotted as a function of best frequency across all 3 auditory stages tested. Dots correspond to individual neurons, and lines correspond to linear fits constrained to have the same slope. (H) Fraction of call-responsive neurons in vMGB, A1 L4, and A1 L2/3 that respond to each call category (*$p < 0.05$, two-sided permutation test with FDR correction). Data underlying this figure can be found in Supporting information file S1 Data. CSD, current source density; FDR, false discovery rate; FS, fast-spiking; GP, guinea pig; LFP, local field potential; RS, regular-spiking.

constructed tonotopic maps (Fig 1C). The location of A1 was confirmed using the direction of the tonotopic gradient and tonotopic reversals. Note that in GPs, the A1 gradient is similar to primates and runs from low frequencies rostrally to high frequencies caudally [6,36,37]. On each track, we also acquired local field potential (LFP) responses to tones at evenly spaced depths, from which we calculated the current source density (CSD) profile of the track (Fig 1B). The thalamorecipient layers (referred to here as A1 L4) were identified based on the presence of a short-latency current sink and LFP polarity reversal [38]. We distinguished between regular-spiking (RS) and fast-spiking (FS) neurons in our recordings using spike width and peak-to-trough amplitude ratio (Fig 1D). About 20% of our recordings were from FS neurons, but call responses were tested in only half these neurons. Only RS neurons are reported in this study. Spontaneous rates of A1 L2/3 neurons (Fig 1E; median: 1.51 spk/s) were not significantly different from A1 L4 neurons (median: 2.31 spk/s) but were significantly lower compared to vMGB neurons (median: 3.67 spk/s; Kruskal–Wallis test $p = 0.008$; post hoc Dunn–Sidak tests vMGB versus A1 L4: $p = 0.1112$, A1 L2/3 versus vMGB: $p = 0.005$, A1 L2/3 versus A1 L4: $p = 0.085$). We sampled over a broad range of neural best frequencies that overlapped with the call frequency range (Fig 1F). Pure tone tuning bandwidths of tone-responsive neurons at all processing stages showed a dependence on best frequency (Fig 1G; ANCOVA with best frequency as covariate, $p = 0.0071$), and after controlling for this frequency dependence, the bandwidths of vMGB neurons were significantly higher than A1 L2/3 neurons (ANCOVA constrained to same slopes; intercept effect $p = 0.0017$; post hoc Tukey honestly significant difference (HSD) vMGB versus A1 L4: $p = 0.053$, A1 L2/3 versus vMGB: $p = 0.0012$). A1 L4 and A1 L2/3 bandwidths were not significantly different ($p = 0.554$). Following basic characterization, we presented a range of GP calls (8 categories, 2 or more exemplars of each category; Fig 2). Note that our vocalization set did not have acoustic power in the 4 to 6 kHz range (Fig 2A), which may explain the relative paucity of call-responsive neurons we encountered in that range, particularly in cortical recordings. All call categories were about evenly represented in neural responses across the processing stages (Fig 1H). The only statistically significant deviations we observed was a small overrepresentation of "Other" calls and a small underrepresentation of "Purr" calls in A1 L2/3 ($p = 0.014$ for both, two-sided permutation test with false discovery rate (FDR) correction for 24 comparisons). All further analyses are based only on call-responsive neurons from the vMGB ($n = 33$), A1 L4 ($n = 67$), and A1 L2/3 ($n = 45$).

## Call selectivity emerges in superficial cortical layers

Call selectivity could emerge through a gradual sharpening of tuning along successive stages of the ascending auditory pathway or could sharply emerge at some processing stage. To distinguish between these models, we quantified the call selectivity of neural populations in vMGB, A1 L4, and A1 L2/3. Fig 3 shows representative examples of neural responses to calls in vMGB (Fig 3A), A1 L4 (Fig 3B), and A1 L2/3 (Fig 3C). Neurons in vMGB and A1 L4 typically responded to many call categories, with responses sustained throughout the call, or occurring at multiple times over the duration of a call. In contrast, neurons in A1 L2/3 responded to very few calls and only for short durations within each call.

Conventionally, response rates and response significance are calculated over a fixed response window, typically encompassing the entire stimulus duration. For a first-pass analysis, we defined selectivity as the number of call categories that, compared to spontaneous rate, evoked a significant response over the entire call duration (1 –highly selective, 8 –no selectivity). The median selectivity of the A1 L2/3 population was 3 call categories, whereas the medians for the A1 L4 and vMGB populations were 6 call categories ($p = 3.5 \times 10^{-6}$; Kruskal–Wallis test). While this approach accurately estimated response properties when response rates were

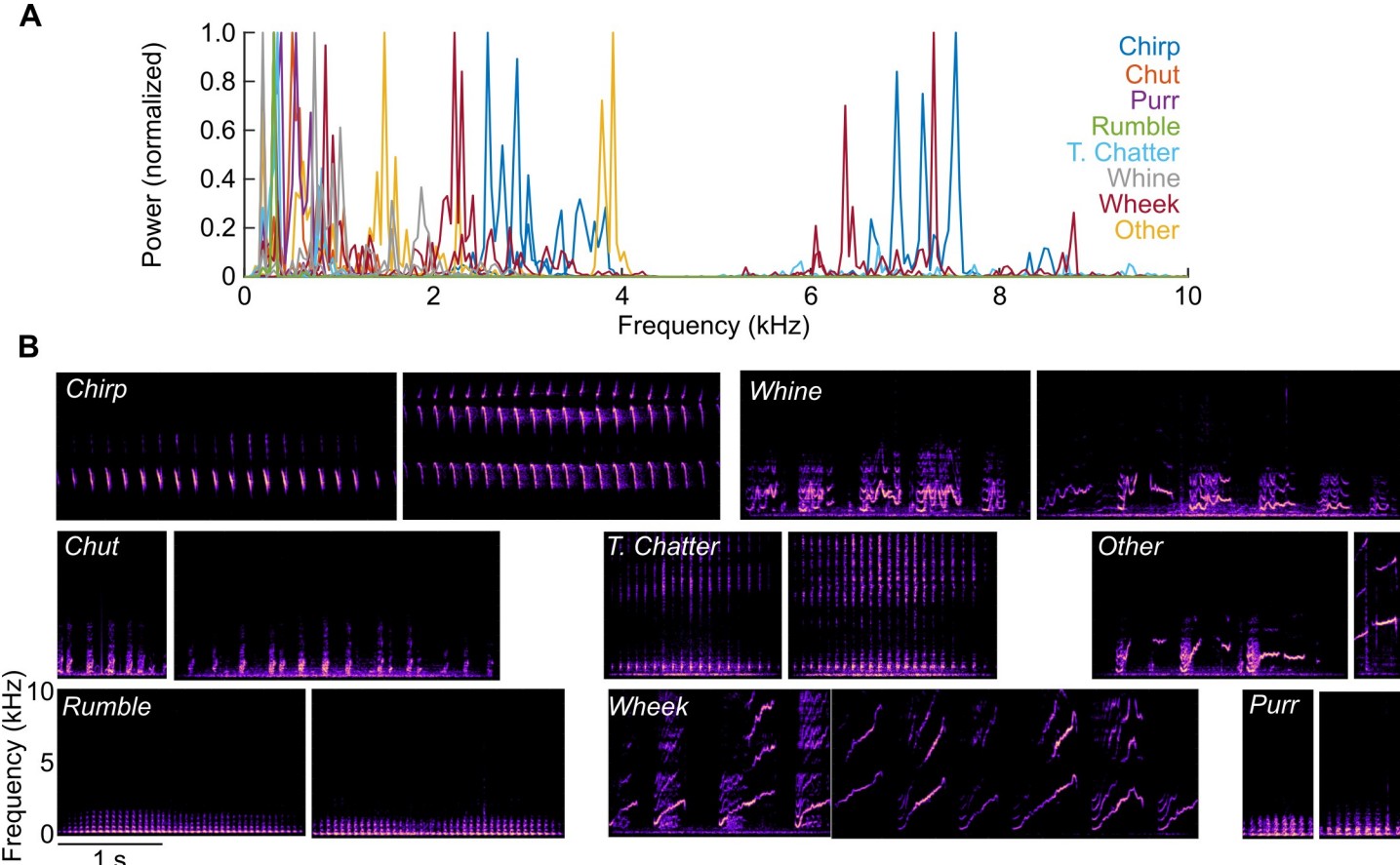

**Fig 2. Spectra and spectrograms of GP calls. (A)** Normalized power spectra of the GP calls used in this study. Colors correspond to different call categories. **(B)** Spectrograms of the GP calls used in this study (8 categories, 2 calls per category). Data underlying this figure can be found in Supporting information file S2 Data. GP, guinea pig.

high and sustained, it sometimes failed to capture feature-selective responses that were restricted to only some time bins of the stimulus, such as those we observed in A1 L2/3. To overcome this limitation, we used an automated procedure to estimate significant response windows for each stimulus (orange boxes in Fig 3; see Materials and methods). If at least one response window was detected for any exemplar belonging to a call category, we conservatively counted the neuron as being responsive to that category.

Over the population of recorded neurons, while vMGB and A1 L4 neurons showed significant responses to most of the categories tested (Fig 4A, left and center; median of 7 categories for both vMGB and A1 L4), nearly a third of A1 L2/3 neurons responded to only 1 or 2 call categories (Fig 4A, right; median = 5). Distributions of call selectivity were not significantly different between the vMGB and A1 L4 populations (medians = 7). In contrast, A1 L2/3 neurons responded to significantly fewer categories of calls ($p = 2.8 \times 10^{-5}$, Kruskal–Wallis test; post hoc Dunn–Sidak corrected $p$-values are as follows: vMGB versus A1 L4: $p = 0.90$; A1 L2/3 versus vMGB: $p = 2.5 \times 10^{-4}$; A1 L2/3 versus A1 L4: $p = 1.9 \times 10^{-4}$). The temporal characteristics of the response and response duration are shown in Fig 4B, where we plot the joint distribution of the number of response windows found per call and the fractional length of call stimuli spanned by response windows in vMGB, A1 L4, and A1 L2/3. While most vMGB and A1 L4 neurons typically exhibited 2 or more response windows per call that spanned a larger fraction

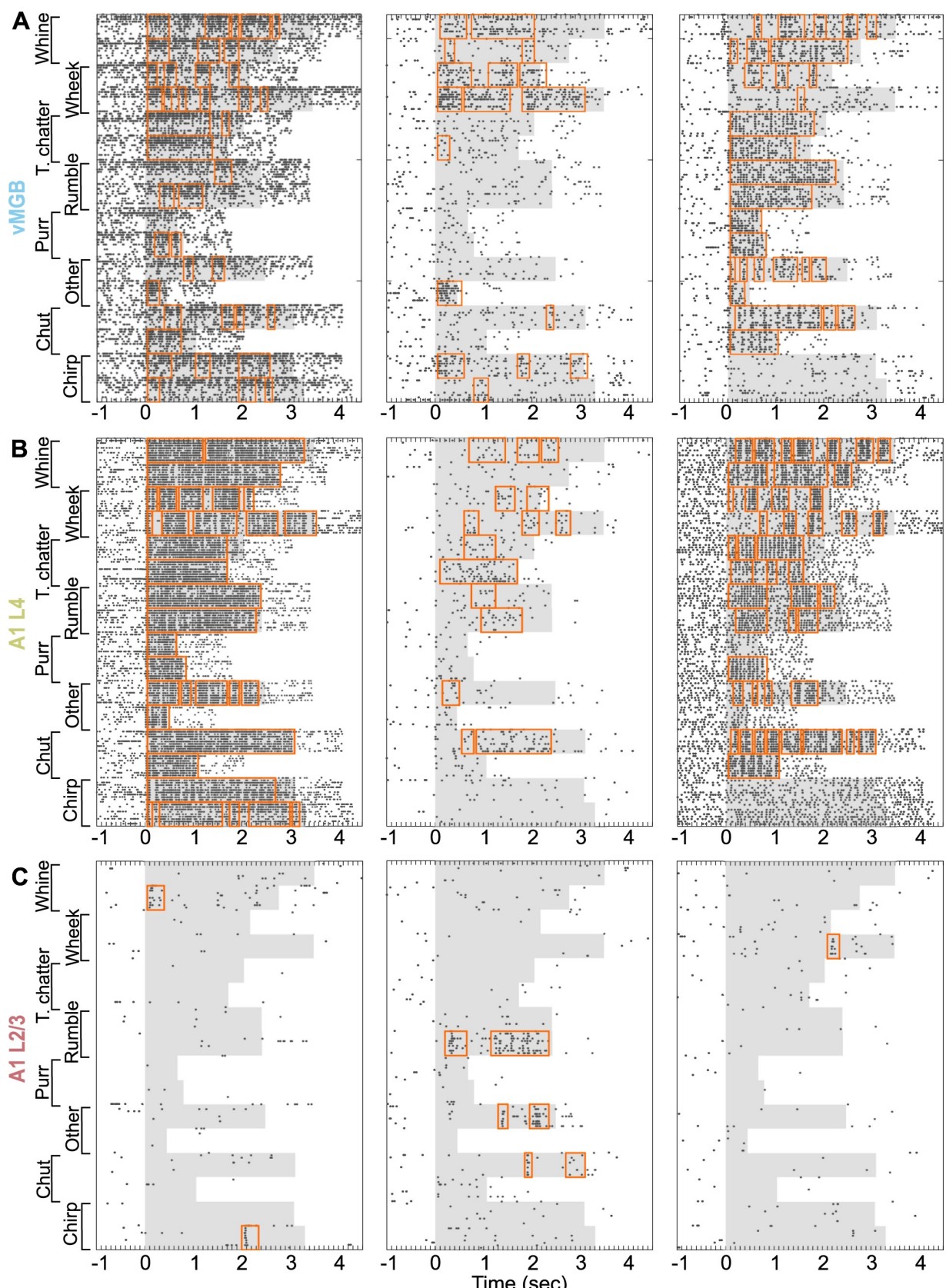

**Fig 3. Detection of response windows.** Spike rasters of 3 call-responsive neurons from **(A)** vMGB, **(B)** A1 L4, and **(C)** A1 L2/3 are plotted. Gray shading indicates stimulus duration, and black dots correspond to spike times. Orange boxes correspond to response windows detected using our algorithm. Data underlying this figure can be found in Supporting information file S3 Data.

of call length, many A1 L2/3 neurons usually exhibited only one response window per call with response windows spanning a smaller fraction of call length. The temporal response characteristics of vMGB and A1 L4 were therefore not significantly different ($p$ = 0.48, 2D Kolmogorov–Smirnov (K–S) [39] test with Bonferroni correction), whereas A1 L2/3 responses were significantly different (A1 L2/3 versus vMGB: $p$ = 0.0008, A1 L2/3 versus A1 L4: $p$ = 0.0023; 2D K–S test with Bonferroni correction). Thus, at the culmination of subcortical processing, vMGB responses are not call selective and in fact mirror earlier studies showing a lack of call selectivity in GP IC [16]. Even at the first cortical processing stage (A1 L4), no transformation to the representation of calls seems to have occurred. However, our data demonstrate that a significant transformation to call representation occurs in many superficial cortical neurons (A1 L2/3). These data strongly support the de novo emergence of call feature-selective responses in the superficial layers of primary auditory cortex.

To evaluate whether neurons specifically responded to only parts of some calls or if neural responses were more evenly distributed across calls using metrics independent of stimulus identity and response window detection parameters, we characterized response sparsity. We defined sparseness as (1) the reduced kurtosis of the trial-wise firing rate distribution and (2) the activity fraction ([40,41]; see Eq 1) of the trial-wise responses. For neurons that responded to most trials about evenly, such as the A1 L4 neuron in Fig 3B (left), the firing rate distribution was approximately normal, resulting in low kurtosis values (Fig 4C, center). In contrast, for neurons that responded strongly only on some trials, and were unresponsive for most trials, such as the A1 L2/3 neuron in Fig 3C (left), the firing rate distribution showed high kurtosis (Fig 4C, right). Over the population, for both sparsity metrics (kurtosis, Fig 4D; and activity fraction, Fig 4E), we found that vMGB and A1 L4 responses were not sparse and not significantly different from each other. Consistent with earlier analyses, compared to both vMGB and A1 L4, A1 L2/3 responses were highly sparse and sparsity distributions were significantly different (Kurtosis: $p$ = $3.2 \times 10^{-5}$, Kruskal–Wallis test; Dunn–Sidak post hoc test $p$-values are as follows: vMGB versus A1 L4: $p$ = 0.99, A1 L2/3 versus vMGB: $p$ = $5.5 \times 10^{-4}$, A1 L2/3 versus A1 L4: $p$ = $1.2 \times 10^{-4}$. Activity fraction: $p$ = $5.2 \times 10^{-4}$, Kruskal–Wallis test; Dunn–Sidak post hoc test $p$-values are as follows: vMGB versus A1 L4: $p$ = 0.79, A1 L2/3 versus vMGB: $p$ = 0.001, A1 L2/3 versus A1 L4: $p$ = 0.004).

These observed differences in A1 L2/3 selectivity and sparsity could not simply be attributed to differences in frequency tuning. As mentioned above, pure tone tuning bandwidths of tone-responsive neurons in A1 L2/3 were not significantly different from A1 L4 neurons (Fig 1G). High call selectivity in A1 L2/3 could also arise if only a few call types are overrepresented in this processing stage. This was not the case in our data—as described earlier, neural preference for call type was about evenly distributed across all tested call types across the processing stages. These controls thus suggest that the emergence of call or feature selectivity in A1 L2/3 is the consequence of cortical computations that result in a meaningful transformation of information representation between processing stages.

Because responses were evoked for more call categories and for larger fractional lengths of the calls in vMGB and A1 L4, and given the overlapping spectral content of call categories that is largely maintained over the call durations (Fig 2), we hypothesized that vMGB and A1 L4 neurons were likely driven by the spectral content of calls, responding when call spectral energy overlapped with the neurons' tone receptive fields. In contrast, despite this overlap of

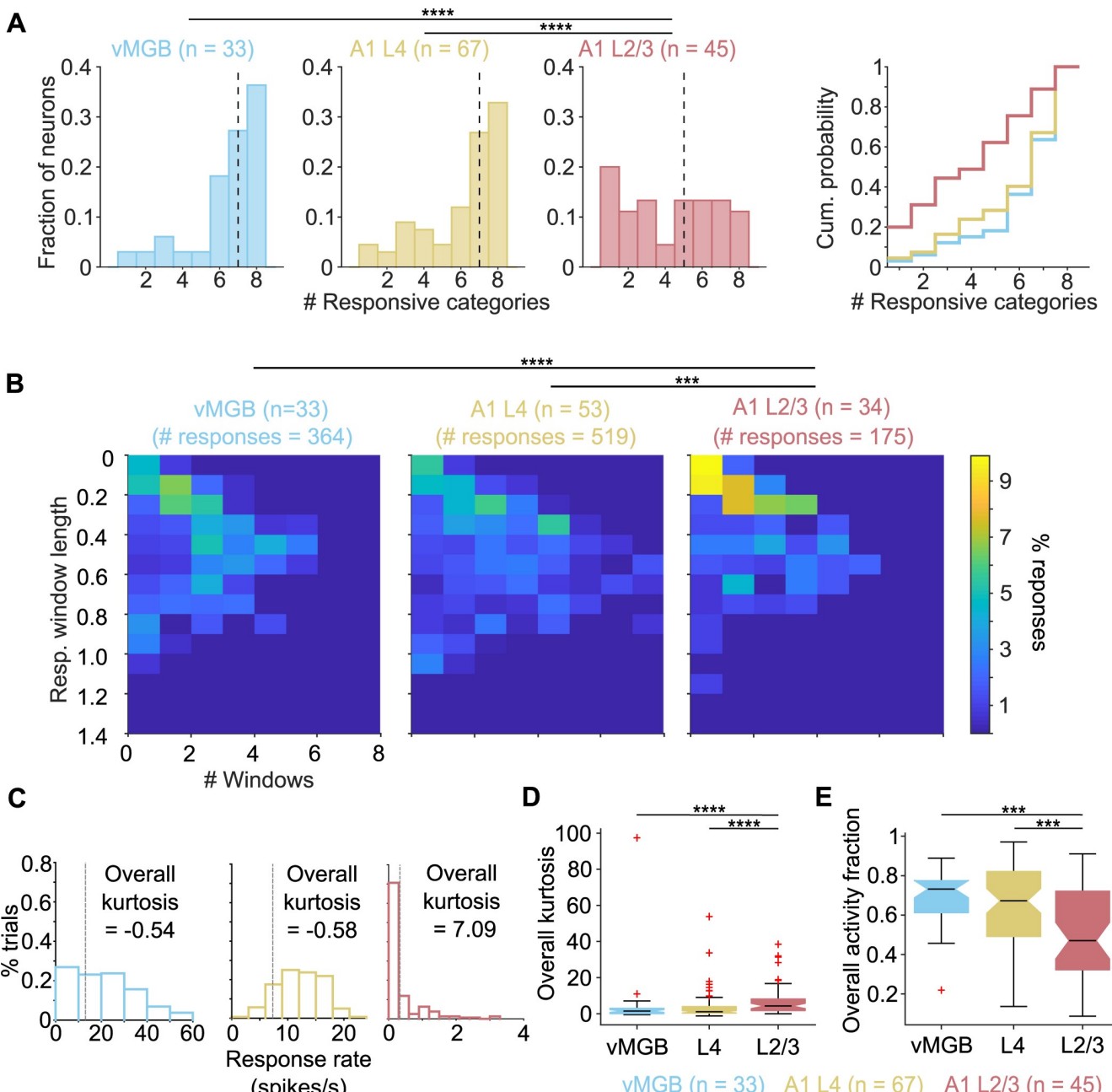

**Fig 4. Neural selectivity for call features emerges in A1 L2/3. (A)** Distributions of call selectivity in vMGB (blue), A1 L4 (yellow), and A1 L2/3 (red). Black dashed lines are medians. Comparison of cumulative distributions is shown on the right. **(B)** Joint distributions of the number of response windows and the fractional length of the call stimuli spanned by all windows exhibited by neurons at the different processing stages. vMGB and A1 L4 neurons tended to exhibit either multiple short windows or a single long window that spanned a large portion of the stimuli. In contrast, A1 L2/3 neurons exhibited 1 or 2 short response windows. **(C)** Distributions of trial-wise response rates in an example vMGB (blue; same neuron as in Fig 3A, left), A1 L4 (yellow; same neuron as in Fig 3B, left), and A1 L2/3 (red; same neuron as in Fig 3C, left) neuron. Kurtosis values calculated over the entire call length are shown. Gray dashed line corresponds to spontaneous rate. **(D)** Distributions of sparseness (kurtosis) across auditory processing stages. A1 L2/3 responses were significantly sparser than A1 L4 and vMGB responses. **(E)** Same as (D) but with activity fraction used as a metric of response sparseness. For all panels except B, Kruskal–Wallis tests with post hoc Dunn–Sidak tests were used for statistical comparisons. For B, a two-dimensional K–S test with Bonferroni correction was used. Asterisks correspond to: $^*p < 0.05$, $^{**}p < 0.01$, $^{***}p < 0.005$, $^{****}p < 0.001$ (exact p-values in main text). Data underlying this figure can be found in Supporting information file S4 Data. K–S, Kolmogorov–Smirnov.

spectral energy across call types, many A1 L2/3 neurons responded to few call types and only in narrow windows, suggesting that they were likely driven by specific spectrotemporal features that occur during calls, consistent with our earlier theoretical model [29]. We tested these hypotheses by estimating the spectrotemporal receptive fields (STRFs) that best explained neural responses across the processing stages.

## Complex spectrotemporal features drive call-selective responses

To determine the call features driving neural responses, we used the Neural Encoding Model System (NEMS [42,43]; https://github.com/LBHB/NEMS) to fit linear–nonlinear (LN) models to neural responses to calls. The input to these models was the concatenated cochleagram of all call stimuli (6 oct. frequency range with 5 steps/oct., 20 ms time bins, approximately 35 seconds total; Fig 5B), constructed using a fast approximation algorithm based on a weighted log-spaced spectrogram and 3 rate-level transformations corresponding to 3 categories of auditory nerve fibers ([44]; https://github.com/monzilur/cochlear_models). A recent study demonstrated that such an input representation adequately captures the auditory input to cortex for the purposes of receptive field estimation [44]. The objective of the encoding model was to estimate a set of linear weights (the STRF of the neuron), which, when convolved with the input cochleagram and then transformed through a point nonlinearity, would yield a predicted peristimulus time histogram (PSTH; Fig 5A; see Materials and methods for details). The correlation coefficient between predicted PSTHs of validation segments of neural responses (labeled $r$ in figures; see Materials and methods) and actual response PSTHs was used as the performance metric. For display and measuring STRF sparsity, we used significance-masked average STRFs (see Materials and methods).

Examples of STRF estimates and comparisons of predicted responses to observed responses are shown for neurons with a range of call selectivities from different subjects in A1 L4 and A1 L2/3 in Figs 5 and 6. For many A1 L4 neurons (Fig 5), STRF estimates that best captured the response showed a clear tuning for specific frequencies, and significant weights were restricted to a narrow range of frequencies and few time bins. While a few call-selective A1 L4 responses could not be directly explained by call energy overlapping with an excitatory receptive field subunit (for example, Fig 5C and 5D), responses of most A1 L4 neurons to calls occurred when call energy was present within the excitatory subunits of the receptive fields (horizontal blue lines in Fig 5E–5H). In contrast, STRFs of A1 L2/3 neurons estimated using the same procedure were often more complex (Fig 6). We observed STRFs with preferences for repetitive features (Fig 6A), harmonically related features (Fig 6G), and frequency-modulated features (Fig 6K). Compared to A1 L4 estimates, significant A1 L2/3 weights spanned a greater range of frequencies and time bins. When we overlaid different stimulus segments on the A1 L2/3 STRFs, we observed that responses did not occur when only stimulus spectral energy matched STRF excitatory subunits (red boxes labeled "3," "4," and "5" in Fig 6). Rather, responses were elicited when complex stimulus features matched multiple STRF subunits (green boxes labeled "1" and "2" in Fig 6).

For example, the unit in Fig 6A–6F showed selective responses to teeth chatter calls, a non-voiced call that contains repetitive pulses of low-frequency energy around 1 kHz accompanied by high-frequency energy around 8 kHz (see spectrogram in Fig 2B). The STRF estimate of this neuron showed excitatory receptive field subunits at approximately 1 kHz and approximately 8 kHz, with an additional excitatory subunit at 8 kHz occurring approximately 100 ms later. Some parts of teeth chatter calls thus closely overlapped the excitatory subunits of STRF, driving strong responses (Fig 6C). But other parts of teeth chatter calls did not drive responses (Fig 6F), possibly because of the faster repetition rate of individual syllables or activity-

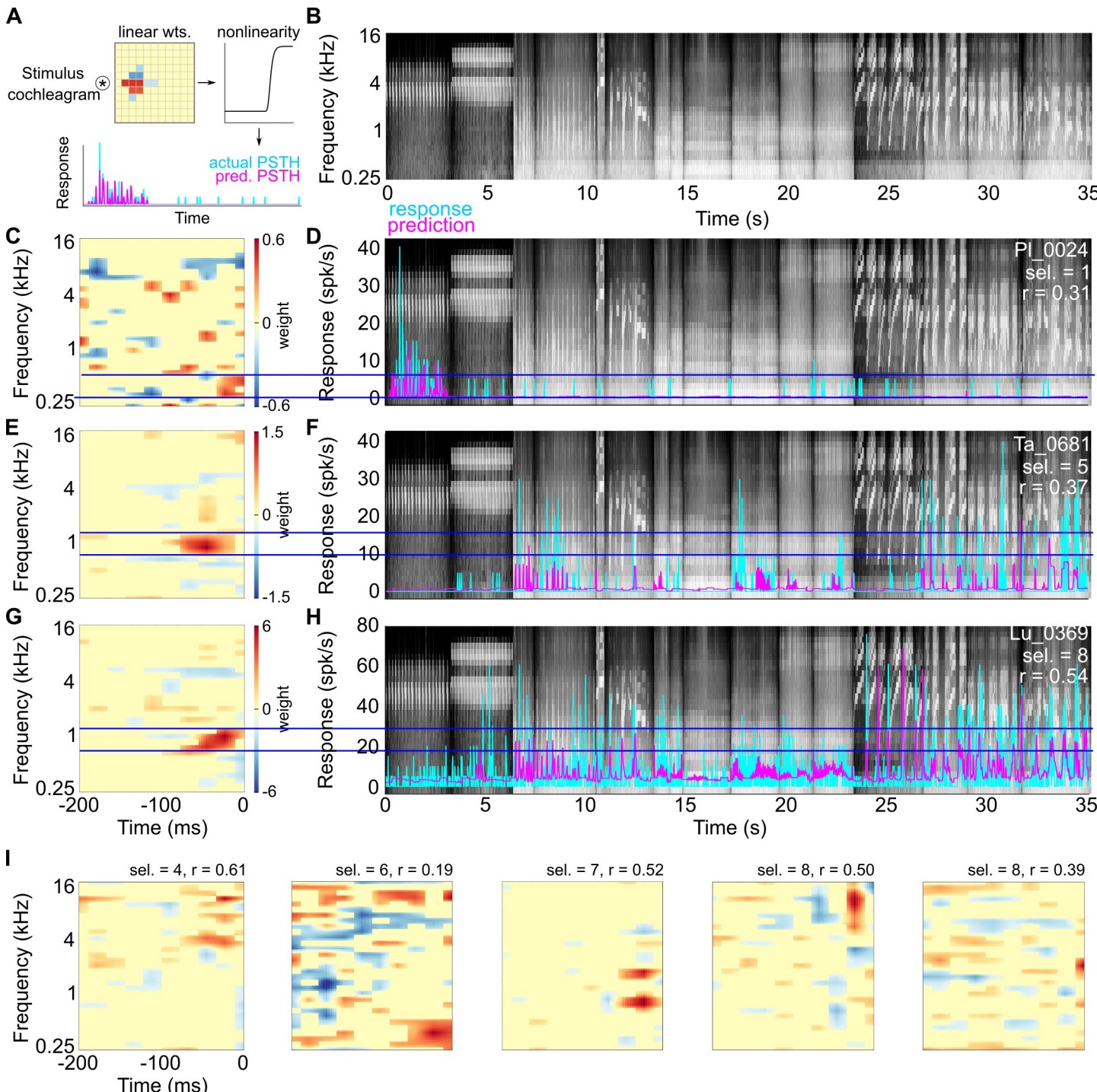

**Fig 5. STRF estimates of example A1 L4 neurons.** (**A**) Schematic of the LN model architecture used to estimate STRFs. (**B**) Stimulus cochleagram of 16 call stimuli (8 categories) used as the input to the model. (**C, E, G**) Mean STRF estimates of 3 A1 L4 neurons with a range of selectivity values. (**D, F, H**) Comparison of predicted PSTHs (magenta) and observed responses (cyan) of these 3 neurons. Horizontal blue lines denote the extent of the frequency tuning of the STRFs. (**I**) Additional examples of A1 L4 STRF estimates (sel. = call selectivity, r = correlation between predicted and actual responses derived from the validation data set). Data underlying this figure can be found in Supporting information file S5 Data. LN, linear–nonlinear; PSTH, peristimulus time histogram; STRF, spectrotemporal receptive field.

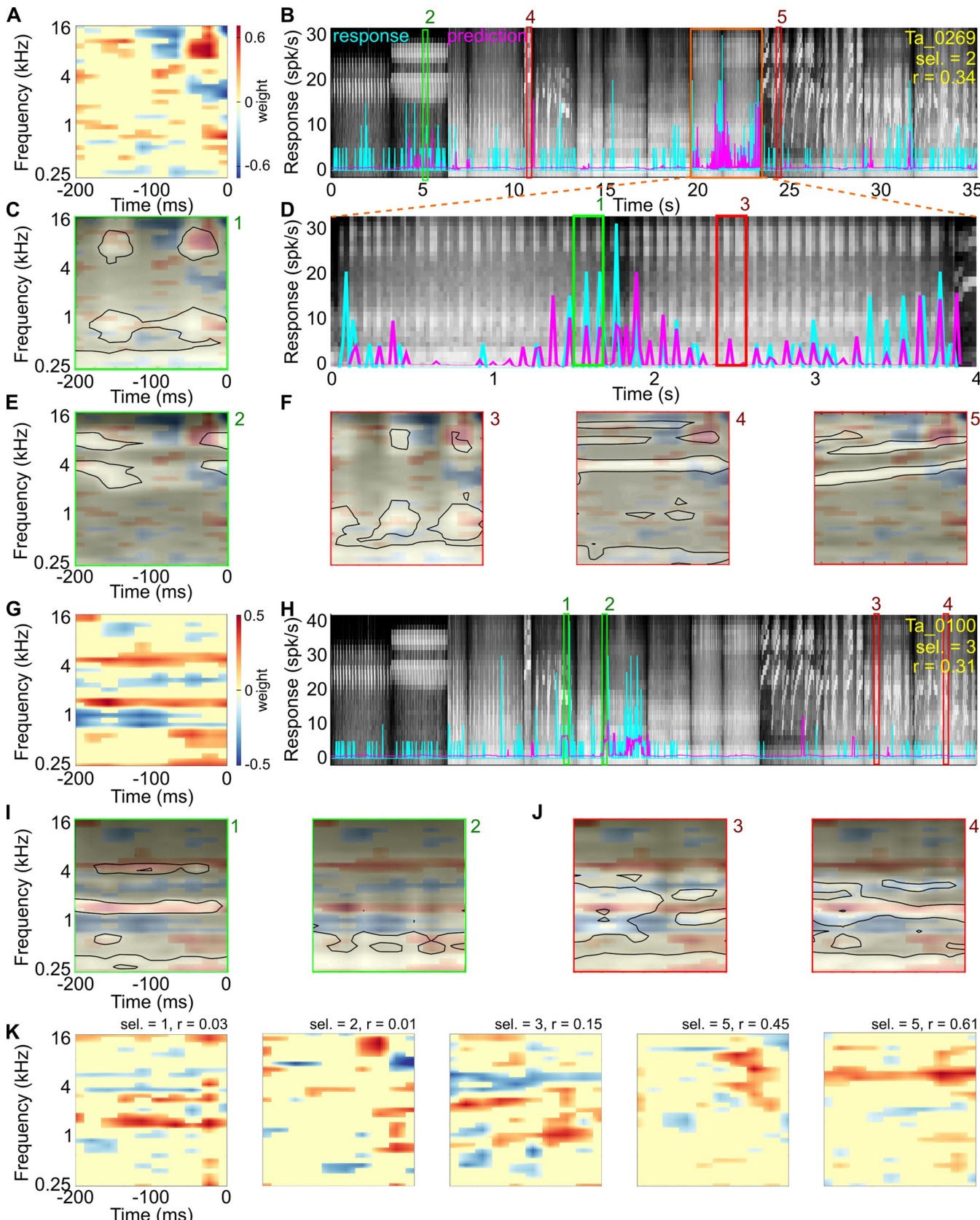

**Fig 6. STRF estimates of example A1 L2/3 neurons. (A, G)** STRF estimates of 2 A1 L2/3 neurons showing complex feature selectivity. **(B, H)** Stimulus cochleagram (background) and comparison of predicted PSTHs (magenta) and observed responses (cyan) of these 2 neurons. **(D)** Expanded cochleagram segment from orange box in B. In B, D, and H, green boxes labeled "1" and "2" correspond to 200 ms long stimulus segments that elicited neural responses. Red boxes labeled "3," "4," and "5" correspond to 200 ms long stimulus segments that did not elicit responses. Numbers correspond to examples shown in panels C, E, F, I, and J. **(C, E, F)** Overlay of stimulus energy in 200 ms long segments corresponding to numbers in B and D (transparency denotes stimulus energy, peak energy is bounded by black contour) on the STRF (colormap) of this unit. **(I, J)** Similar to C, E, and F but for the other A1 L2/3 example. **(K)** Additional examples of complex STRFs of A1 L2/3 neurons. Data underlying this figure can be found in Supporting information file S6 Data. PSTH, peristimulus time histogram; STRF, spectrotemporal receptive field.

dependent adaptation of spiking activity. A second call exemplar that had repetitive energy at 8 kHz (a chirp call) also drove responses in this neuron to a lesser extent (Fig 6E), but other vocalizations with 8 kHz energy that did not have a repetitive structure did not drive responses (e.g., wheek calls; Fig 6F). A second example unit that required the presence of harmonic structure is shown in Fig 6G–6J. This unit appeared to require at least 2 of the excitatory STRF subunits to be activated to produce a response. The selectivity for multiple frequency components in this unit was reminiscent of "harmonic template neurons" that have been reported in marmoset auditory cortex [45]. This unit responded even when different frequency combinations were excited by different calls (Fig 6I), underscoring the intuition that these units could not be described as a simple spectral filter. Fig 6K shows further examples of STRF estimates of units that showed selective responses to call features.

Over the population of neurons, we did not find significant differences in the performance of the LN models to fit training data segments from vMGB, A1 L4, or A1 L/3 neurons (Fig 7A, left; $p = 0.684$, Kruskal–Wallis test), suggesting that the model converged to a solution similarly across the 3 processing stages. However, while the LN models generalized to the validation data segments with similar performance in vMGB and A1 L4 neurons, generalization was significantly worse for A1 L2/3 neurons (Fig 7A, right; Kruskal–Wallis test, $p = 0.0003$; Dunn–Sidak post hoc test $p$-values are as follows: vMGB versus A1 L4: $p = 0.999$, A1 L2/3 versus vMGB: $p = 0.003$, A1 L2/3 versus A1 L4: $p = 0.0006$). Critically, model generalization performance was correlated with call selectivity across all processing stages (Fig 7B; ANCOVA with

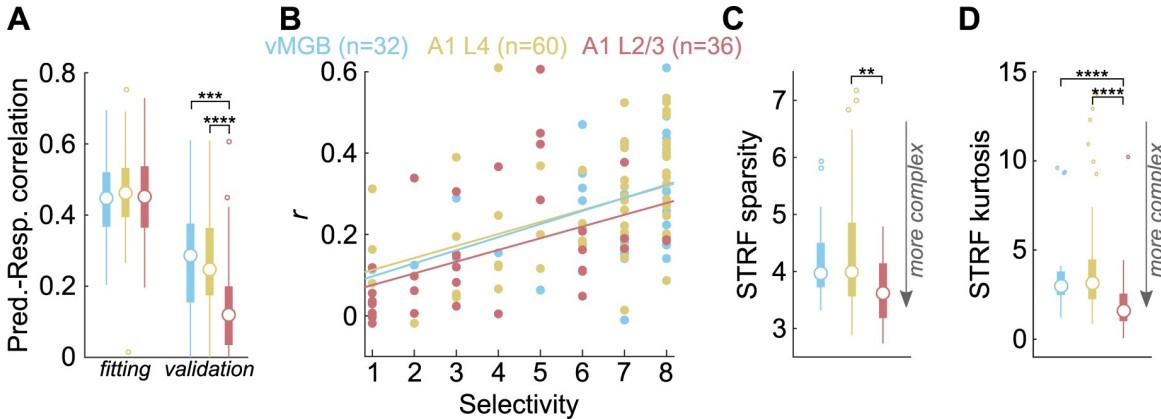

**Fig 7. Performance and complexity of STRF estimates across processing stages. (A)** Performance of LN models on test and validation data from MGB (blue), A1 L4 (yellow), and A1 L2/3 (red). Discs denote medians, thick lines denote interquartile range, and thin lines correspond to the extent of the distribution. Outliers are shown as dots. **(B)** Model validation performance plotted as a function of call selectivity across processing stages. Dots are individual neurons, and lines correspond to linear fit. **(C)** Distributions of STRF sparsity across processing stages. Colors and symbols as earlier. **(D)** Distributions of STRF kurtosis across processing stages. Colors and symbols as earlier. For all panels except B, Kruskal–Wallis tests with post hoc Dunn–Sidak tests were used for statistical comparisons. For B, an ANCOVA with selectivity as a covariate was used. Asterisks correspond to: $^*p < 0.05$, $^{**}p < 0.01$, $^{***}p < 0.005$, $^{****}p < 0.001$ (exact $p$-values in main text). Data underlying this figure can be found in Supporting information file S7 Data. LN, linear–nonlinear; STRF, spectrotemporal receptive field.

selectivity as covariate; $p = 2.83 \times 10^{-7}$). We note that several neurons with a call selectivity of 1 showed very low and nonsignificant $r$ values. These observations suggest that more complex and nonlinear models may be required to capture these highly selective responses.

We used 2 metrics to compare the complexity of STRF structure across processing stages. First, we used STRF sparsity, defined as the maximum absolute value of the significance-masked STRF divided by the standard deviation of the significance-masked STRF [46,47]. For "simple" STRFs, the maximum value would be high, whereas standard deviation would be low, resulting in high STRF sparsity values. For complex STRFs where many weight values are large, the maximum value and standard deviation would be comparable, resulting in lower STRF sparsity values (Fig 7C; Kruskal–Wallis test, $p = 0.008$), with post hoc tests revealing a significant difference between A1 L2/3 and A1 L4 neurons (Dunn–Sidak post hoc test, $p = 0.006$). As a second metric, we quantified the kurtosis of STRF weight values (after significance masking). STRFs with simple structure would show weight distributions with high kurtosis, with most of the weights concentrated in 1 or 2 subunits, and the rest of the weights equaling zero. Complex STRFs would be expected to have a more normal distribution of weight values. We found a significant effect of processing stage on kurtosis (Fig 7D; Kruskal–Wallis test, $p = 3.3 \times 10^{-5}$), with Dunn–Sidak post hoc tests revealing significant differences between A1 L2/3 and vMGB ($p = 0.0007$) as well as between A1 L2/3 and A1 L4 ($p = 0.0001$). These statistical results were qualitatively unchanged even when neurons with nonsignificant $r$ values were excluded. These observations supported our hypothesis that whereas vMGB and A1 L4 neurons responded to call stimuli in a manner that was largely consistent with their spectral tuning properties, A1 L2/3 neurons were driven by more complex spectrotemporal features present in calls that could not be well fit by linear models.

## Emergence of call feature selectivity in A1 L2/3 confers high stimulus-specific information on to individual A1 L2/3 neural responses

While our data show that A1 L2/3 neurons become call selective by restricting their responses to specific call features, the consequence of this emergence of call selectivity on decoding call identity from A1 L2/3 neural activity is unclear. An obvious expectation would be that increasing the feature selectivity of single neurons would result in unique activity patterns in response to some calls, thereby leading to higher information carried by these neurons about call identity. Conventionally, MI [48] has been used to estimate the amount of information about stimulus identity carried by neural responses [49–52]. Intuitively, for our call stimulus set consisting of 16 calls, a neuron that exhibits 16 unique response patterns, each corresponding to a call, would provide the maximal MI about the stimulus set (in this case, 4 bits of information). We computed the MI between the responses and stimuli, limiting our analyses to the first 1,457 ms of response and postresponse period (see Materials and methods). When we computed the average MI in 100 ms time bins (50 ms slide; see Materials and methods) of the population of A1 L4 neurons as has been done in most earlier studies [49–52], we found low information levels throughout the response duration (Fig 8A, yellow) that were not significantly different (two-sided $t$ test with FDR correction at each time point) from population MI present in the vMGB population (Fig 8A, blue). However, consistent with a recent result showing decreasing information content in the ascending auditory pathway of anesthetized GPs [53], we found significantly lower MI levels in the A1 L2/3 population (Fig 8A, red). We confirmed that this result held over a wide range of window sizes used for analysis (S1 Fig and data in Supporting information file S9 Data).

To understand how lower population MI levels might arise and to test whether this negatively impacted stimulus decodability in A1 L2/3, we decomposed how information was

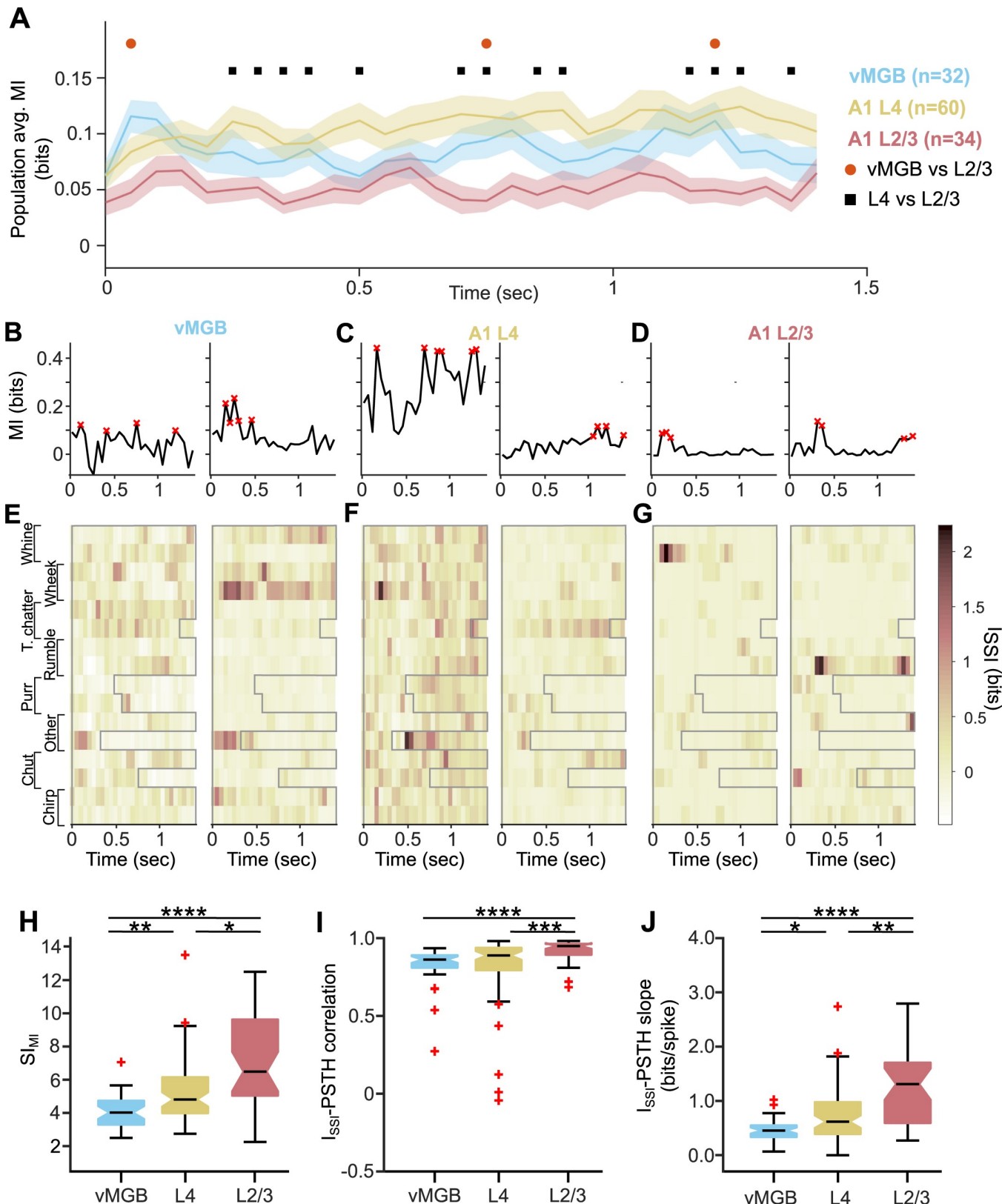

**Fig 8. Reformatting of stimulus information in A1 L2/3. (A)** Population average of MI as a function of time in vMGB (blue), A1 L4 (yellow), and A1 L2/3 (red) neurons. Lines correspond to means and shading to 1 SEM. Colored dots represent results of statistical testing ($p < 0.05$; two-sided $t$ test with FDR

correction for multiple comparisons). **(B–D)** MI for 2 example neurons each from vMGB (B), A1 L4 (C), and A1 L2/3 (D). The example neurons are the same as the left 2 examples from Fig 3A–3C. Red crosses correspond to high MI time bins. **(E–G)** $I_{SSI}$ for the vMGB neurons in (B), the A1 L4 neurons in (C), and the A1 L2/3 neurons in (D). Darker colors correspond to higher $I_{SSI}$ values. **(H)** Distributions of $SI_{MI}$ for vMGB, A1 L4, and A1 L2/3 neurons. Horizontal line corresponds to median, and colored area corresponds to interquartile range. **(I)** Distributions of $I_{SSI}$–PSTH correlation coefficients for vMGB, A1 L4, and A1 L2/3 neurons. **(J)** Distributions of $I_{SSI}$–PSTH slopes for vMGB, A1 L4, and A1 L2/3 neurons. Asterisks correspond to: $^*p < 0.05$, $^{**}p < 0.01$, $^{***}p < 0.005$, $^{****}p < 0.001$ (Kruskal–Wallis test with post hoc Dunn–Sidak tests, exact $p$-values in main text). Data underlying this figure can be found in Supporting information file S8 Data. FDR, false discovery rate; MI, mutual information; PSTH, peristimulus time histogram.

distributed across 2 factors, (1) individual neurons and (2) individual stimuli, in the vMGB, A1 L4, and A1 L2/3 neural populations. First, we examined how MI was distributed over the individual neurons that make up the population average in Fig 8A. Fig 8C shows MI as a function of time for 2 example A1 L4 neurons (the same neurons as in Fig 3B, left and center). Although the magnitudes of MI are different, the MI over time is sustained in both cases, which means that when averaged, the mean MI will also be sustained over time (as in Fig 8A, yellow). In contrast, Fig 8D shows MI for 2 example A1 L2/3 neurons (the same neurons as in Fig 3C, left and center). Here, the MI is close to zero for many time bins and shows peaks in time bins that are nonoverlapping between neurons, which means when averaged, the mean MI will be at a low value (as in Fig 8A, red). Second, we decomposed the MI into stimulus-specific information ($I_{SSI}$; [54–56]), which measures how much information about each stimulus is provided by the response. Note that the conventionally computed MI is the weighted average of $I_{SSI}$ across all stimuli. Fig 8E–8G show the decomposition of the MI of the example neurons in Fig 8B–8D, respectively, into the $I_{SSI}$ for each call stimulus. In A1 L4 (Fig 8F), $I_{SSI}$ was evenly distributed across all stimuli and time bins, resulting in the average (the MI; Fig 8C) being at a sustained level over time. In A1 L2/3, however (Fig 8G), $I_{SSI}$ was very high (approaching 3 bits) for specific stimuli only at specific time bins. Thus, average $I_{SSI}$ across stimuli, as is done to compute MI (Fig 8D), approached zero for most time bins and severely underestimated the informativeness of the response.

To quantify whether a high-MI time bin (see Materials and methods; red crosses in Fig 8B–8D) arises from an approximately normal distribution of $I_{SSI}$ across all stimuli for that time bin (as in Fig 8F), or from a highly skewed $I_{SSI}$ distribution across stimuli for that time bin (as in Fig 8G), we computed an MI sparsity index. For this analysis, we first identified high-MI time bins (bins that have more than mean + 1 standard deviation of population MI) and quantified the distribution of $I_{SSI}$ values only for these bins ($SI_{MI}$; see Materials and methods). $SI_{MI}$ increased significantly between all 3 processing stages tested (Fig 8H; $p = 1.1 \times 10^{-6}$, Kruskal–Wallis test; Dunn–Sidak post hoc test $p$-values are as follows: vMGB versus A1 L4: $p = 0.007$, A1 L2/3 versus vMGB: $p = 5.0 \times 10^{-7}$, A1 L2/3 versus A1 L4: $p = 0.012$), with A1 L2/3 neurons being informative about only a few calls in their most informative time bins.

MI analysis only takes into account spike patterns but does not distinguish between the presence or absence of spikes. In other words, if a neuron responds to 15 of the 16 call stimuli and is inhibited by 1 call, the information provided by this neuron about the stimulus set is equivalent to that provided by a neuron that responds to only one call. To determine whether $I_{SSI}$ is provided by the presence or absence of spikes, we computed the cross-correlation between the PSTH and $I_{SSI}$ across all time bins for neurons in vMGB, A1 L4, and A1 L2/3. Compared to vMGB and A1 L4, A1 L2/3 neurons showed higher $I_{SSI}$–PSTH correlations, suggesting that A1 L2/3 responses were informative because of presence of spikes (Fig 8I; $p = 9.3 \times 10^{-6}$, Kruskal–Wallis test; Dunn–Sidak post hoc test $p$-values are as follows: vMGB versus A1 L4: $p = 0.240$, A1 L2/3 versus vMGB: $p = 1.0 \times 10^{-5}$, A1 L2/3 versus A1 L4: $p = 0.001$). Compared to A1 L4, the $I_{SSI}$–PSTH relationship in A1 L2/3 also showed a significantly higher slope, indicating that each spike from an A1 L2/3 neuron carried greater

stimulus-specific information ([Fig 8J]; $p = 2.7 \times 10^{-6}$, Kruskal–Wallis test; Dunn–Sidak post hoc test $p$-values are as follows: vMGB versus A1 L4: $p = 0.021$, A1 L2/3 versus vMGB: $p = 1.3 \times 10^{-6}$, A1 L2/3 versus A1 L4: $p = 0.007$). We confirmed that these results were consistent over a wide range of window sizes used for analysis ([S1 Fig]).

[Table 1] is a summary of all statistical comparisons of basic tuning properties, selectivity metrics, STRF metrics, and information theoretic metrics of vMGB, A1 L4, and A1 L2/3 neurons. Where possible, we estimated effect size using the nonparametric Cliff delta ($d$) ([57]; range: [−1, 1]; implemented using code from https://github.com/GRousselet/matlab_stats). As a guideline for interpreting these values, the effect size may be considered "small" if $0.11 < |d| \leq 0.28$, "medium" if $0.28 < |d| \leq 0.43$, and "large" if $|d| > 0.43$ [58]. Asterisks denote statistical significance. If call selectivity gradually developed over the 3 processing stages, one would expect to see differences in selectivity parameters pairwise between all 3 processing stages. In contrast, if selectivity arose de novo in superficial cortical layers, vMGB and A1 L4 parameter distributions would not be significantly different (second column), but A1 L2/3 and A1 L4 (as well as A1 L2/3 versus vMGB; third and fourth columns) would show significant differences. Our results support the latter possibility and the idea that while subcortical activity and inputs to A1 represent vocalizations densely and based on spectral content, a call feature-based representation emerges in A1 L2/3 that dramatically transforms how information about conspecific calls is represented in A1 outputs.

**Table 1. Statistical summary of comparisons between vMGB, A1 L4, and A1 L2/3.**

| Parameter | vMGB vs. A1 L4 | A1 L2/3 vs. vMGB | A1 L2/3 vs. A1 L4 |
|---|---|---|---|
| ***Basic properties*** | | | |
| Bandwidth ANCOVA (post hoc Tukey HSD) | 0.36 (*) | −0.54 (***) | −0.12 (n.s.) |
| Spontaneous rate | 0.25 (n.s.) | −0.5 (***) | −0.24 (n.s.) |
| ***Selectivity parameters*** | | | |
| Selectivity (overall firing rate) | −0.14 (n.s.) | −0.32 (****) | −0.48 (****) |
| Selectivity (response windows) | 0.07 (n.s.) | −0.52 (****) | −0.44 (****) |
| No. of windows and response length 2D K–S (Bonferroni correction) | (n.s.) | (****) | (***) |
| Kurtosis | −0.04 (n.s.) | 0.49 (****) | 0.46 (****) |
| Activity fraction | 0.10 (n.s.) | −0.47 (***) | −0.36 (***) |
| ***STRF parameters*** | | | |
| $r$ | 0.02 (n.s.) | −0.46 (***) | −0.45 (****) |
| STRF sparsity | −0.13 (n.s.) | −0.28 (n.s.) | −0.38 (**) |
| STRF kurtosis | −0.01 (n.s.) | −0.54 (****) | −0.51 (****) |
| ***MI analyses (100 ms time bins)*** | | | |
| Population MI 2-sided $t$ test (FDR correction) | n.s. | Few time points | Many time points |
| $SI_{MI}$ | −0.44 (**) | 0.65 (****) | 0.40 (*) |
| $I_{SSI}$–PSTH correlation | −0.21 (n.s.) | 0.66 (****) | 0.43 (***) |
| $I_{SSI}$–PSTH slope | −0.36 (*) | 0.67 (****) | 0.39 (**) |

Numbers denote effect size (Cliff delta, range [−1, 1]). Asterisks in parentheses denote statistical significance (****$p < 0.001$, ***$p < 0.005$, **$p < 0.01$, *$p < 0.05$, n.s., not significant). All tests are Kruskal–Wallis tests with post hoc Dunn–Sidak tests unless noted otherwise.

FDR, false discovery rate; HSD, honestly significant difference; K–S, Kolmogorov–Smirnov; MI, mutual information; PSTH, peristimulus time histogram; STRF, spectrotemporal receptive field.

## Discussion

Although many previous studies have explored the neural representation of conspecific calls in subcortical and cortical areas across species [6–9,15–27,59], exactly where and how call selective responses emerge in the auditory processing hierarchy has remained unclear. In mice, some studies have suggested that selectivity for ultrasonic vocalizations (USVs) in a manner not consistent with spectral content might arise at subcortical stations [5] and lead to an over-representation of USV-selective responses in the IC [60]. However, other studies have suggested that this overrepresentation is explained by a tonotopic expansion of the representation of those frequencies and that USV responses are in fact consistent with spectral tuning of neurons [61]. In bats, the majority of neurons in subcortical processing stations responded to calls consistent with neurons' frequency tuning [3,62]. In GPs, single neurons in the IC are not selective for particular call types or call features [16]. In the MGB, although single neurons follow call envelopes less precisely [15] and neural responses to calls are less predictable from neurons' tone tuning [63], responses do not differentiate between natural and reversed versions of calls [64], suggesting that MGB responses are not call or call feature selective. At the level of A1, some studies have reported that single neurons show selectivity for natural calls over reversed calls [18] or that neurons seem to respond to calls that share similar spectrotemporal features [23], but by and large, neural responses to calls seem to be explained by the frequency tuning of neurons [7,21]. At the level of secondary cortex, neurons have been shown to be highly selective for call type in primates [8,9] and GPs (Area S and VRB [6]). However, because of some technical limitations of these studies, including the use of anesthesia, limited stimulus sets, multiunit recordings, or not comparing across processing stages, specifically across cortical laminae, it is difficult to evaluate where transformations to call representation begin to occur. Answering the "where" question is a critical first step that will enable the targeting of experiments probing the neural mechanisms underlying these transformations to the appropriate target processing stage. In this study, we overcame these limitations by simultaneously (1) conducting experiments in unanesthetized animals, (2) using an extensive set of conspecific calls as stimuli, (3) comparing across thalamic and cortical processing stages, and (4) separating A1 neurons recorded from thalamorecipient and superficial layers. We found that whereas call representations in vMGB and A1 L4 were similar, a critical transformation occurs between A1 L4 and A1 L2/3. While vMGB and A1 L4 neurons seemed to respond primarily to the spectral content of calls resulting in a dense representation of calls, many A1 L2/3 responses were contingent on the presence of specific spectrotemporal features, resulting in a highly sparse representation of calls.

This observed transformation is consistent with previously reported increases in the nonlinearity of neural receptive fields in marmosets [31], increases in sparsity of responses in rats [65], and some reports of increased receptive field complexity in superficial A1 layers (in cats; [66–68]). This transformation is also consistent with ultrahigh field human fMRI studies showing that supragranular BOLD responses are less readily explained using simple frequency tuning models [33]. Thus, the transformation of sound representation between A1 L4 and A1 L2/3 appears to be a conserved phenomenon across species, from GPs to humans. In nonhuman primates, secondary auditory cortical areas have been shown to exhibit call-selective responses [8,9], and the highest sensory cortical regions of the auditory processing pathway preferentially represent conspecific calls [10–12]. Our results suggest that the emergence of call feature selectivity at supragranular A1 layers is a critical first step in building call-selective cortical specializations.

How could highly feature-selective neurons be generated? In an earlier study in marmosets, many A1 neurons recorded at shallow cortical depths were combination selective, i.e., these

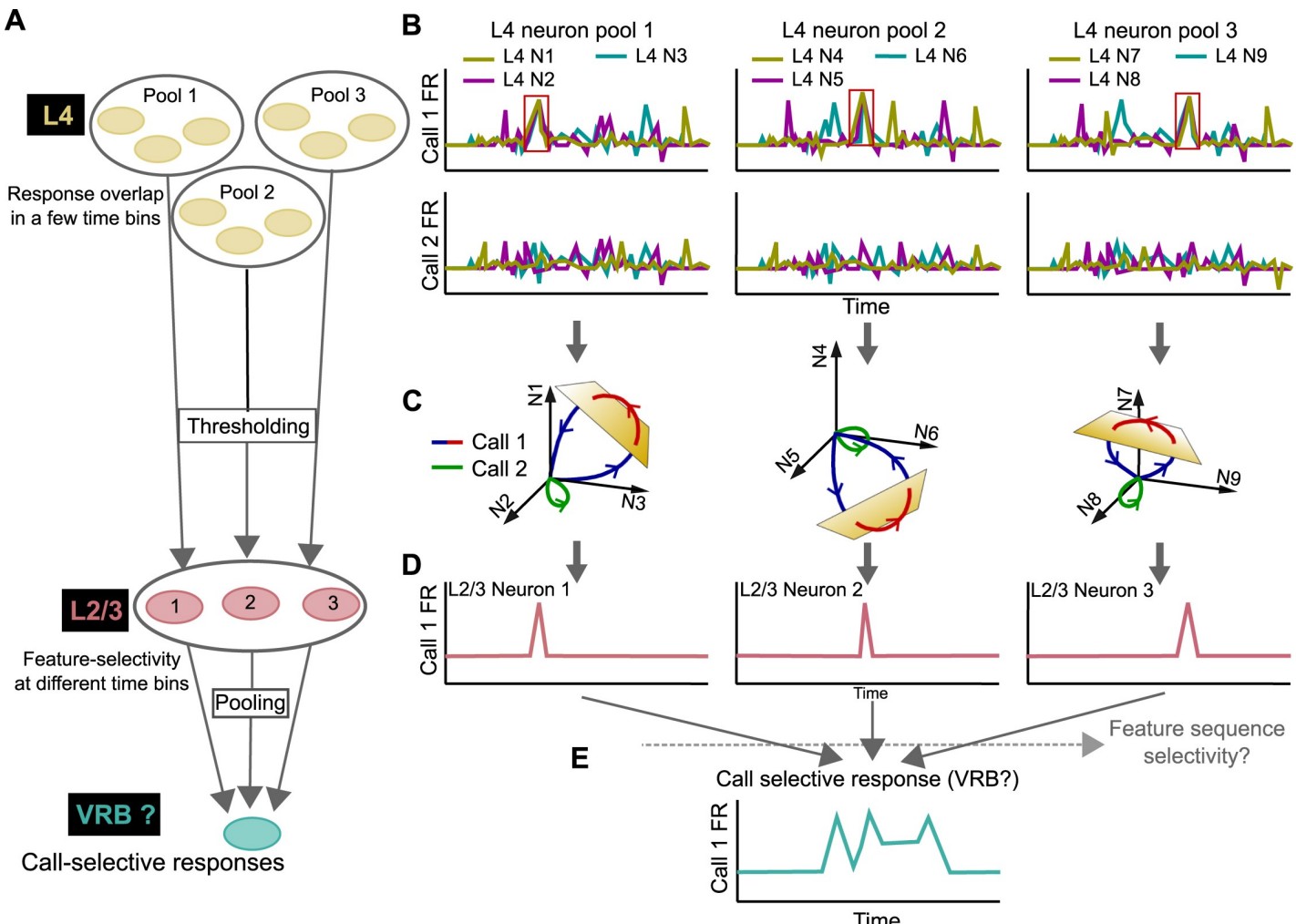

**Fig 9. Working model of generating call selectivity in the auditory cortical hierarchy. (A)** Alternating nonlinear (high threshold) and more linear (pooling) stages that could result in call-selective responses in secondary cortex. **(B)** Schematic of nonselective A1 L4 neural responses that could show overlap in a few time bins. **(C, D)** A high threshold could be applied to different pools of A1 L4 neurons to result in A1 L2/3 responses that are selective to specific call features. **(E)** A more linear operation could pool over A1 L2/3 neurons that are selective for features belonging to the same call category to result in call category-selective secondary cortical responses.

neurons showed responses only when specific frequencies were present with precise temporal relationships [31]. Such nonlinear mechanisms could generate call feature selectivity, but how precise temporal delays necessary for this computation are generated in the A1 circuit remains an open question. A second possibility is that although A1 L4 neurons are not selective for call type in that they respond to all categories, responses to some calls in a subset of time bins may be marginally stronger (Fig 9A and 9B). Pooling a number of A1 L4 neurons that exhibit similar marginally stronger responses to the same time bin, but whose responses are uncorrelated otherwise, could accentuate the differences between this preferred time bin and other bins. The higher $SI_{MI}$ observed in A1 L4 neurons compared to vMGB neurons supports the notion that there may be local periods of high information in the A1 L4 population responses. Applying a strong nonlinearity to these pooled inputs could in principle create A1 L2/3 responses that are highly selective for particular spectrotemporal call features (Fig 9C and 9D). Supporting the notion that A1 L2/3 neurons might be applying high thresholds is the fact that A1 L2/3

neurons are known to exhibit very low spontaneous rates across species [31,63], including in our own data (Fig 1E).

Extending this model to the next linear (pooling) stage, the responses of multiple feature-selective A1 L2/3 neurons that respond to features belonging to the same call category could be integrated by neurons in secondary cortical areas to result in sustained call category-selective responses (Fig 9E). In anesthetized GPs, neurons that show dense firing with high contrast between call categories, which is highly useful in discriminating between call categories, have been reported in secondary areas VRB and S [6]. It is yet to be determined whether additional mechanisms could be used to increase call category selectivity by further restricting responses to only if call features are detected in a particular temporal sequence, which, for example, could be achieved by some forms of dendritic computation [69–71], or using local axonal conduction delays at very short time scales [72]. Our proposed model is based on model architectures with alternating linear and nonlinear stages that have been used to explain responses in inferotemporal cortex [73]. These models are based on exclusively excitatory and feedforward operations. Other models, for example, incorporating recurrent excitatory inputs that have been shown to sharpen cortical tuning [74], or those involving cortical inhibition that could also fine-tune cortical selectivity [75–78], represent alternative architectures that are more complex but biologically realistic. Specific cortical inhibitory cell types, for example, somatostatin-expressing interneurons, might play a role in generating sharp frequency tuning [79]. Thus, extensive theoretical and experimental work is necessary to test these models and dissect the neural mechanisms underlying the generation of feature selectivity.

What could be the advantages of a highly sparse representation? Extensive work in the visual cortex has proposed that sparse coding could allow for increased storage capacity for associative memory, is more energy efficient, and could make read out by downstream areas easier [80]. The possibility of easier readout is especially interesting in the auditory system, where highly variable continuous inputs need to be parsed and sequenced into categorical units (for example, words in human speech or call category in animal communication calls). The "dense" codes we found in vMGB and A1 L4 are redundant to some degree because neurons respond to highly overlapping stimulus sets. Thus, the activity of a single neuron in A1 L4 signaled the presence of multiple call features, with the actual feature identity being encoded over the population. This is reflected in our information theoretic analysis showing that in A1 L4, MI is distributed both over time bins and over neurons. A1 L2/3 effectively decorrelated A1 L4 activity, so that single neurons now carried high levels of information about the stimulus. One consequence of this decorrelation is an increase in the dimensionality of sound representation, which could serve to "untangle" [81] highly variable representations of different sound categories. As mentioned earlier, in a further processing step, a linear pooling operation could be used to pool responses of A1 L2/3 neurons that respond to different features of the same call type, resulting in truly call category-selective responses such as those observed in secondary cortical areas [8,9]. Further analysis is necessary to quantify the dimensionality of sound representation in different cortical layers and the separability of different call categories. In the auditory system, a second consideration for a neural code is robustness to environmental noise—realistic listening conditions add reverberations, noise, and competing sounds to the target sound impinging on our ears. It remains to be seen whether the feature-selective responses we have observed in A1 L2/3 neurons will remain invariant to these perturbations and will provide a more robust representation of sounds than the dense representations in A1 L4.

In conclusion, by recording from successive auditory processing stages in awake animals using a rich and behaviorally relevant stimulus set, we have demonstrated that rather than a gradual emergence of feature selectivity over the auditory processing hierarchy, selectivity for

sound features appears to emerge de novo in the superficial layers of auditory cortex, resulting in a highly sparse representation of sounds by A1 L2/3 neurons. Our data thus identify that critical transformations to sound representations occur at the superficial layers of A1. These data set the stage for further studies investigating the biophysical and circuit mechanisms by which call feature selectivity arises from nonselective inputs, and how these feature-selective responses could be read out by downstream call category-selective neurons. Our data suggest that the root of observed cortical specializations for call processing [10–12] could in fact reside in primary auditory cortex.

## Materials and methods

### Ethics

All experimental procedures conformed to NIH Guide for the care and use of laboratory animals and were approved by the Institutional Animal Care and Use Committee (IACUC) of the University of Pittsburgh (protocol number 18062947).

### Animals

We acquired data from 4 male and 2 female adult, wild-type, pigmented GPs (*Cavia porcellus*; Elm Hill Labs, Chelmsford, Massachusetts), weighing approximately 600–1,000 g over the course of the experiments.

### Surgical procedures

All experiments were conducted in unanesthetized, head-fixed, passively listening animals. To achieve head fixation, a custom head post was first surgically anchored onto the skull using dental acrylic (Metabond, Parkell, Brentwood, NY) following aseptic techniques under isoflurane anesthesia. Chambers for electrophysiological recordings were positioned over the location of auditory cortex using anatomical landmarks [6,36,37]. Postsurgical care, including administration of systemic and topical analgesics, was provided for 3 to 5 days. Following a 2-week recovery period, animals were gradually adapted to the experimental setup by slowly increasing the duration of head fixation.

### Acoustic stimuli

All stimuli were generated in Matlab (Mathworks, Natick, MA) at a sampling rate of 100 kHz, converted to analog (National Instruments, Austin, TX), attenuated (TDT, Alachua, FL), power amplified (TDT, Alachua, FL), and delivered through a speaker (TangBand, Taipei, Taiwan) located approximately 90 cm from the animal on the contralateral side. We used a wide variety of stimuli including pure tones, noise bursts, frequency- and amplitude-modulated sounds, two-tone pips, and conspecific vocalizations as search stimuli to initially detect and isolate single units. Once we isolated a unit, we delivered pure tones (50 or 100 ms) covering 7 octaves in frequency (200 Hz to 25.6 kHz, 10 steps/oct.) at different sound levels (20 dB SPL spacing) to characterize its frequency response area. We defined the best frequency of the unit as the frequency eliciting the highest firing rate, best level as the sound level eliciting the highest firing rate. The bandwidth of the unit was estimated using a rectangle fit to the frequency tuning curve at the best level [82]. After characterizing basic tuning properties, we presented conspecific vocalization stimuli. All vocalizations were recorded in our animal colony using Sound Analysis Pro [83] by placing one or more animals in a sound-attenuated booth and by recording vocalizations using a directional microphone (Behringer, Willich, Germany). Two observers manually segmented and classified vocalizations into categories based on

previously published criteria [6,34,35]. We verified high interobserver reliability using Cohen Kappa statistic (κ = 0.8). In electrophysiological experiments, we typically presented 2 exemplars each of 8 vocalization categories (16 vocalization stimuli; 0.4- to 3.5-second length depending on call type; typically, 10 repetitions of each stimulus). For some units, we presented additional exemplars belonging to some categories (24 stimuli) but only presented 5 repetitions. All vocalizations were normalized for RMS power and presented at 70 dB SPL in random order, with a random intertrial interval between 2 and 3 seconds. For some units, we also presented vocalizations to which we added reverberations or noise (not presented in the current manuscript).

## Electrophysiology

All recordings were conducted in a sound-attenuated booth (IAC, Naperville, IL) whose walls were lined with anechoic foam (Pinta Acoustics, Minneapolis, MN). Animals were head fixed in a custom acrylic enclosure affixed to a vibration–isolation tabletop that provided loose restraint of the body. We recorded the activity of single units in the vMGB and identified cortical laminae of primary auditory cortex (A1). We sequentially performed small craniotomies (approximately 1 mm dia.) within the recording chamber using a dental drill (Osada, Los Angeles, CA) attached to a stereotactic manipulator (David Kopf Instruments, Tujunga, CA) to reach regions of interest. For vMGB recordings, we targeted previously published stereotactic coordinates [84,85] by performing a caudally angled craniotomy in the caudal part of the chamber. The location of the electrode in the vMGB was confirmed using electrophysiological properties (strong tone responses, low response latency, and expected tonotopic organization [86,87]). For cortical recordings, we performed craniotomies over the expected anatomical location of A1 [6,36,37] angled to be roughly perpendicular to the cortical surface. We used strong tone responses and tonotopic reversals to confirm that the recording location was within A1. In each recording session, we used a hydraulic microdrive (FHC) to advance a tungsten microelectrode (FHC, Bowdoin, ME or A-M Systems, Sequim, WA; 2 to 5 MΩ impedance) through the dura into the underlying target tissue. Electrophysiological signals were digitized and amplified using a low-noise amplifier (Ripple Neuro, Salt Lake City, UT), and data visualized online (Trellis software suite). We played a wide variety of search stimuli while slowly advancing the electrode. When a putative spike was detected, we used a template-matching algorithm for online spike sorting to isolate single units. Sorting was further refined offline at the conclusion of the experimental session (MKSort, provided by Ripple Neuro, Salt Lake City, UT). Using this technique, we typically acquired spike data from 1 to 3 single units simultaneously. Spike waveforms were classified into putative RS and FS categories using the peak-to-trough ratio and spike width as parameters. We only considered well-isolated single units, defined as having a peak amplitude at least 5.5 standard deviations above noise baseline, for further analysis. For A1 recordings, we sequentially recorded neural activity from superficial to deep cortical layers. At the end of each electrode track, we advanced the electrode to a depth of approximately 2 mm and acquired LFP responses every 100 μm while retracting the electrode. To do so, we presented 100 repetitions of a pure tone at 70 dB SPL, with pure tone frequency chosen to match the best frequency of the recorded column. From these local potential data, we calculated the CSD, defined as the second spatial derivative of the LFP, based on which we assigned recorded units to thalamorecipient or superficial layers [38]. After the electrode was completely retracted, the craniotomy was filled with antibiotic ointment, and recording chambers sealed using a silicone polymer (KwikSil or similar). Recording sessions were limited to 4 hours, and we typically recorded from each craniotomy for 4 to 8 electrode tracks. Craniotomies were sealed with dental cement after data acquisition was completed.

## Data analysis and statistics

Analysis was based on data from 45 L2/3 RS neurons, 67 L4 RS neurons, and 33 vMGB neurons that responded to at least one vocalization in our stimulus set. We also isolated 10 call-responsive FS neurons from A1 recordings, which were not analyzed in this study.

**Response window analysis.**   We obtained response rate estimates limited to small time bins using an algorithm similar to Issa and Wang [88] (also see [89–91]). Briefly, we started with seed windows selected using relaxed criteria and gradually added additional windows until the final window met stringent criteria. To do so, we first determined whether the responses to any call in any 100 ms window (50 ms slide) located from 50 ms poststimulus onset until 100 ms poststimulus offset met 2 criteria: (1) the average rate exceeded 6 SEM of the spontaneous rate and (2) the trial-wise response distribution within the window was significantly different from the spontaneous response distribution with $p_{soft} \leq 0.1$ (single-tailed $t$ test with FDR correction; this test is used for determining all $p$-values for response window analysis). The initial window could then grow in either direction by adding neighboring windows, if (1) the response in window to be added met a threshold of $p_{soft} \leq 0.1$, (2) average rate in the enlarged window exceeded 10 SEM of the spontaneous rate, and (3) the trial-wise response distribution within the enlarged window met a threshold of $p_{add} \leq 0.01$. We successively added response segments until these thresholds could not be met. To avoid a single bursty trial from spuriously increasing response rate, we replaced trial-wise rates with a z-score $> 1.96$ by the mean response rate of the enlarged window. The resultant window was considered the final response window if (1) the average rate exceeded 14 SEM of the spontaneous rate, (2) if the trial-wise response distribution within the final window met a threshold of $p_{final} \leq 0.0001$, and (3) if responses were present on at least 60% of the trials. Any windows less than 100 ms apart were coalesced if the resulting window still met the 3 final stringent criteria. If no response windows were detected for any call, we relaxed the following parameters in order: minimum trial threshold decreased to 50%, z-score for burst detection increased to 2.5, and window length increased to 200 ms (slide = 100 ms). For example, minimum responsive trial threshold was decreased to 50%, and burst detection z-score increased to 2.5 for the neuron in Fig 3C (right). Parameters for automated response window analysis were initially chosen to broadly match response regions to visual judgements of 3 independent observers in a small sample of neurons from the 3 processing stages. Results were verified to be largely consistent over a range of parameter values. While this automated analysis reliably detected excitatory responses, because of the very low spontaneous rates of cortical neurons, inhibitory responses could not be captured. Thus, when responses were mainly inhibitory rather than excitatory (2 neurons in A1 L2/3 and 9 neurons in A1 L4), the number of calls with significant responses was determined manually by 3 independent observers.

**Quantification of selective responses.**   We quantified the selectivity of neural responses based on the following metrics: (1) We defined call selectivity as the number of call categories with significant responses—if at least one response window was detected for any exemplar belonging to a category, we counted the neuron as being responsive to that category. (2) The number of response windows per call. (3) The length of the response, which was the sum of all window lengths within a call, expressed as a fraction of the total length of that call. Together, metrics (2) and (3) indicated if a neuron was feature selective—for highly feature-selective neurons, we observed a small number of short windows, whereas for neurons with low selectivity, we observed many short windows or a single long window. We compared selectivity across processing stages using Kruskal–Wallis tests followed by pairwise post hoc tests. To quantify differences in feature selectivity across processing stages, we constructed two-dimensional distributions of the number of windows versus window length and evaluated significance using 2D K–S tests [38] with Bonferroni correction.

**Sparsity.**  We estimated sparsity using 2 metrics. (1) As the reduced kurtosis of trial-wise firing rate responses [82,92], computed over a single window from 50 ms after stimulus onset to 100 ms after stimulus offset. A reduced kurtosis of zero indicates a normal distribution of firing rates across stimuli or response bins, suggesting a response that is not feature selective. High kurtosis values arise when many response rates are zero and few response rates are high, suggesting highly feature-selective responses. (2) As the activity fraction [40,41], defined as:

$$A = \frac{\left[\sum_{i=1}^{N} r_i/N\right]^2}{\sum_{i=1}^{N} [r_i^2/N]} \tag{1}$$

An activity fraction close to zero signifies highly sparse responses, and activity fraction close to one signifies dense responses. Sparsity across processing stages was compared using Kruskal–Wallis tests followed by pairwise post hoc Dunn–Sidak tests.

**Receptive field models.**  We used the NEMS ([42,43]; https://github.com/LBHB/NEMS) as a platform to build LN models and estimate STRFs of call-responsive neurons. The input to the model consisted of the cochleagram of all call stimuli concatenated in time. To compute the cochleagram, we used a fast approximation algorithm that used weighted log-spaced frequency bins and 3 rate-level transformations corresponding to 3 categories of auditory nerve fibers ([44]; https://github.com/monzilur/cochlear_models). Previous work has shown that this transformation can adequately capture the inputs to auditory cortex [44]. The resolution of the cochleagram was set at 5 steps/oct. in frequency (total 6 oct. spanning 250 Hz to 16 kHz) and 20 ms in time. Linear weights and the parameters of a point nonlinearity (double exponential function) were estimated by gradient descent to minimize the squared error between the predicted PSTH and the actual PSTH (computed in 20 ms bins, averaged over 10 repetitions). The matrix of linear weights was taken to represent the receptive field or STRF of the neuron. We performed a nested cross-validation, where for every neuron's call responses, we used 90% of the data to fit the models and the remaining 10% to validate the models. This procedure was repeated 10 times using nonoverlapping segments of validation data to fit and test the model, yielding 10 STRF estimates. The correlation coefficient between predicted responses from the validation data set ($r$) and actual responses was used as a metric of goodness of fit. A bootstrap procedure was used to test for significance of $r$ values. For quantifying STRF complexity and display, we used the mean STRF (over the 10 cross-validation runs) multiplied by a significance mask. To estimate the mask, we used a bootstrap procedure by scrambling the actual linear weight matrices 1,000 times to estimate the distribution of weights at each time and frequency bin and used a two-tailed permutation test to evaluate if the observed STRF mean weights differed significantly (using FDR correction for 310 comparisons) from the bootstrap distributions. To quantify the complexity of STRFs, we used STRF sparsity [46,47], defined as the maximum absolute value of the significance-masked STRF divided by the standard deviation of the significance-masked STRF, and as a second metric, the kurtosis of significance-masked STRF weights. Sparsity and kurtosis across processing stages were compared using Kruskal–Wallis tests followed by Dunn–Sidak post hoc tests.

**Information theoretic analyses.**  We used stimulus-specific information ($I_{SSI}$) [54–56] to estimate the amount of information that each recorded neuron provided about each stimulus. We also computed the weighted average of $I_{SSI}$ across stimuli to determine overall information content, which is conventionally referred to as the MI between the stimulus and response. Only neurons that had completed 10 trials for all the stimuli were considered for this analysis. Intuitively, if a neuron shows a consistent response pattern to a given stimulus, then it has high $I_{SSI}$ about that stimulus. To quantify $I_{SSI}$, we extracted responses beginning 50 ms before stimulus onset and lasting until 50 ms after the length of the longest stimulus [49] in windows

of varying lengths (14, 50, 100, 200, 300, and 400 ms with slide equal to half the window size). Because vocalizations had different lengths, based on a previous study [77], we restricted these analyses to the first 1,457 ms of the responses (457 ms shortest call length followed by a 1,000 ms poststimulus period where a subsequent stimulus was guaranteed not to be present). Responses to longer calls were thus truncated. For each window size, the $I_{SSI}$ in each time bin was calculated as:

$$I_{SSI} = \sum_{resp} p(resp|stim) * I_{SP}(resp) \qquad (2)$$

where $I_{SP}(resp)$ is the information conveyed by a specific response pattern, calculated as:

$$I_{SP}(resp) = TotalEntropy - ConditionalEntropy(resp) \qquad (3)$$

$$TotalEntropy = -\sum_{stim} p(stim) * log_2(p(stim)) \qquad (4)$$

$$ConditionalEntropy(resp) = -\sum_{stim} p(stim|resp) * log_2(p(stim|resp)) \qquad (5)$$

To correct for estimation bias arising from finite trial numbers that likely undersample response probability distributions, we subtracted an all-way shuffled estimate of $I_{SSI}$ (average of 100 randomizations [56]) from the value of $I_{SSI}$ estimated earlier. All reported values refer to the bias-corrected $I_{SSI}$ estimate.

Having obtained these $I_{SSI}$ estimates, we computed how $I_{SSI}$ values were distributed across time bins and across stimuli and how $I_{SSI}$ correlated with spiking responses of each neuron. To quantify how $I_{SSI}$ values were distributed across time bins and across stimuli, for each window size, we calculated an MI sparsity index ($SI_{MI}$), defined as the mean kurtosis of $I_{SSI}$ values in high-MI time bins, with high-MI bins defined as bins with MI values exceeding 1 standard deviation of the MI values across all time bins. To determine whether high $I_{SSI}$ resulted from the presence or absence of spiking, we calculated the correlation between the $I_{SSI}$ and PSTH. Finally, to determine how much information was conveyed by each spike, we determined the slope of the $I_{SSI}$ versus PSTH distribution. Distributions of information-theoretic measures between A1 L4 and A1 L2/3 were compared using Kruskal–Wallis tests with post hoc pairwise tests. We chose the 100 ms window size (50 ms slide) for all comparisons shown in the main manuscript. Similar results were obtained across most tested window sizes (S1 Fig and S9 Data).

## Supporting information

**S1 Fig. Mutual information analyses performed across a range of analysis window sizes.**
(PDF)

**S1 Data. Data underlying Fig 1.**
(XLSX)

**S2 Data. Data underlying Fig 2.**
(XLSX)

**S3 Data. Data underlying Fig 3.**
(XLSX)

**S4 Data. Data underlying Fig 4.**
(XLSX)

**S5 Data. Data underlying Fig 5.**
(XLSX)

**S6 Data. Data underlying Fig 6.**
(XLSX)

**S7 Data. Data underlying Fig 7.**
(XLSX)

**S8 Data. Data underlying Fig 8.**
(XLSX)

**S9 Data. Data underlying S1 Fig.**
(XLSX)

## Acknowledgments

We thank Dr. Yi Zhou (ASU) for insightful comments on the manuscript. We thank Isha Kumbam and Samuel Li for recording and classifying guinea pig vocalizations; Dr. Marianny Pernia, Shi Tong Liu, and Dr. Flora Antunes for assistance with electrophysiological experiments; and Dr. Marianny Pernia for assistance with analysis. We thank Stacy Cashman and Mark Petts for surgical support; Dr. Amanda Fisher for veterinary support; and Jillian Harr, Sarah Gray, Julia Skrinjar, Brent Barbe, and Elizabeth Chasky for animal care.

## Author Contributions

**Conceptualization:** Srivatsun Sadagopan.

**Formal analysis:** Manaswini Kar, Stephen V. David, Srivatsun Sadagopan.

**Funding acquisition:** Srivatsun Sadagopan.

**Investigation:** Pilar Montes-Lourido, Srivatsun Sadagopan.

**Resources:** Stephen V. David.

**Software:** Stephen V. David, Srivatsun Sadagopan.

**Supervision:** Srivatsun Sadagopan.

**Visualization:** Pilar Montes-Lourido, Manaswini Kar.

**Writing – original draft:** Manaswini Kar, Srivatsun Sadagopan.

**Writing – review & editing:** Manaswini Kar, Srivatsun Sadagopan.

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
