## [Editor Report · Decision Letter 0]

25 Sep 2020

Dear Dr Sadagopan, 

Thank you for submitting your manuscript entitled "Abrupt emergence of vocalization selectivity in primary auditory cortex" for consideration as a Discovery Report by PLOS Biology.

Your manuscript has now been evaluated by the PLOS Biology editorial staff as well as by an academic editor with relevant expertise and I am writing to let you know that we would like to send your submission out for external peer review.

Also, after discussing your submission with the other members of the editorial team, we think your study would be a better fit as a Short Report rather than a Discovery Report, given that aspects of your study are supportive of previous theoretical work and that hierarchical changes in neuronal call selectivity have been previously described. We therefore request that when revising your manuscript you submit it as a Short Report. 

Like Discovery reports, Short Reports can be up to 4 figures and you should not need to do any reformatting at this stage. Short reports may be based on a small number of experiments that might not completely flesh out the biological phenomenon under study. We aim for our Short Reports to be provocative and of general interest, in such a way as to spur future research, and/or to present a concise set of clever experiments that reconcile previously conflicting observations, resolve a specific conundrum, or simply apply elegant techniques to elucidate a brief answer to an interesting scientific question. More information on the Short Report format can be found here: https://journals.plos.org/plosbiology/article?id=10.1371/journal.pbio.3000248

I would also like to mention that we would still be happy to consider follow up studies, if the advance is enough.

Please re-submit your manuscript within two working days, i.e. by Sep 29 2020 11:59PM.

Kind regards,

Lucas Smith, Ph.D.,

Associate Editor

PLOS Biology

---

## [Decision Letter · Decision Letter 1]

12 Nov 2020

Dear Dr Sadagopan,

Thank you very much for submitting your manuscript "Abrupt emergence of vocalization selectivity in primary auditory cortex" for consideration as a Short Reports at PLOS Biology. Your manuscript has been evaluated by the PLOS Biology editors, an Academic Editor with relevant expertise, and by independent reviewers.

The reviews of your manuscript are appended below. You will see that the reviewers find the work potentially interesting. However, based on their specific comments and following discussion with the academic editor, I regret that we cannot accept the current version of the manuscript for publication. The reviewers suggest that the study does not currently provide sufficient proof that there is, indeed, an “abrupt transition” between L4 and L2/3 – with Reviewer 2 suggesting that there are large differences between MGB and L4 responses, and discrepancies between results in Fig 1 and 3 that reflect this. Reviewer 3 also arguing that the differences between vMGB, L4 and L2/3 are not as straightforward as suggested. In addition, the reviewers feel that the study would be strengthened with data added to support the speculation that L2/3 responses are contingent on the presence of specific spectrotemporal features.

We remain interested in your study and we would be willing to consider resubmission of a comprehensively revised version of the manuscript that thoroughly addresses all the reviewers' comments and strengthens the conclusions of the study with new data and analysis. We cannot make any decision about publication until we have seen the revised manuscript and your response to the reviewers' comments. Your revised manuscript would be sent for further evaluation by the reviewers.

We appreciate that these requests represent a great deal of extra work, and we are willing to relax our standard revision time to allow you six months to revise your manuscript. We expect to receive your revised manuscript within 6 months, however please email us (plosbiology@plos.org) if you have any questions or concerns, or would like to request an extension. At this stage, your manuscript remains formally under active consideration at our journal; please notify us by email if you do not intend to submit a revision so that we may end consideration of the manuscript at PLOS Biology.

**IMPORTANT - SUBMITTING YOUR REVISION**

*Resubmission Checklist*

*Published Peer Review*

*PLOS Data Policy*

*Blot and Gel Data Policy*

Sincerely,

Lucas Smith, Ph.D.,

Associate Editor,

lsmith@plos.org,

PLOS Biology

REVIEWS:

Reviewer's Responses to Questions

PLOS authors have the option to publish the peer review history of their article (what does this mean?). If published, this will include your full peer review and any attached files.

Reviewer #1: Yes: Michelle Moerel

Reviewer #2: No

Reviewer #3: No

Reviewer #1: The authors record responses to vocalizations in three stages of the auditory processing hierarchy of the guinea pig: ventral MGB, L4 of primary auditory cortex (A1), and L2/3 of A1. Regular spiking neurons in ventral MGB and L4 of A1 responded to many calls for an extended duration of time, while L2/3 A1 neurons were much more selective (both in call type and in response timing). The authors interpret these findings as evidence for an abrupt change in the sound representation between L4 and L2/3 of A1. Specifically, they suggest that neurons in L2/3 of A1 selectively respond to informative call features (i.e., spectrotemporal segments of calls present in category examplars), which could be a first step to call category-selective neurons.

The research is highly relevant, as it is so far unclear how the category selectivity present in higher order auditory cortex (across species) emerges from the faithful representation of sound acoustics in lower processing stages. The findings of this study clearly show where in the auditory pathway an important representational change happens, and could serve as basis for follow up studies into the mechanism underlying this change.

Comments:

1. Abstract: Is "… optimized for complete representations of sounds" the best phrasing? Neurons are also optimized for the complete representation of a sound if they are category selective. Perhaps "neurons faithfully reflect acoustic input"?

2. Abstract: the use of the term feature-selectivity is confusing (4th sentence), as it could be taken to mean simple tonotopic tuning (i.e., frequency is an acoustic feature after all).

3. Caption of Figure 2: should this say "call-responsive" instead of "call-selective" neurons? The first neuron in A responds to all 8 call categories, so this neuron is not call-selective.

4. The analyses that underlie Figures 3B-C combine two effects: that of call-selectivity and the response duration per call. That is, it does not allow discriminating if regions differ from each other because neurons respond to fewer calls, or respond shorter (or a combination of both). It would be interesting to restrict these analyses to sounds for which there is at least one responsive window

5. Fig. 3D: how can the y-axis (resp. window length [fraction of call length]) go higher than 1? Are there neurons that respond longer than the sound duration?

6. Results, page 11: "These data suggest that vMGB and A1 L4 neurons are likely driven by the frequency content of calls, when call spectral energy overlaps with the neurons' tone receptive fields. Consequently, vMGB and A1 L4 neurons show poor call selectivity because call spectral energy remains more or less consistent over the call duration and is similar across GP call types. In contrast, despite this overlap of spectral energy across call types, A1 L2/3 neurons respond only in narrow windows because they are likely driven by specific spectrotemporal features that occur during calls, consistent with an earlier theoretical model [29]." This is a relevant and interesting conclusion. However, the reason for poor selectivity in MGB/A1 and reason for the narrow response windows in A1 L2/3 is speculation; this is not supported by the data. Is it possible to better support this statement by the data? For example, by estimating RFs (showing the emergence of combination selectivity in L2/3), and/or by providing more information on the acoustics of the calls and relating that to the neural best frequency + call responsivity?

7. Results, page 13: "… could not be attributed to low-level differences in tuning properties". Perhaps rephrase, as it is not clear what is meant by "low-level differences". After all, differences in spectrotemporal tuning may also be termed low-level. Instead, the authors seem to intend to refer here exclusively to frequency tuning.

8. Figure 4: Why is data from the ventral MGB missing from the MI analyses? How does it compare to A1?

9. Discussion: "While vMGB and A1 L4 neurons seemed to respond primarily to the spectral content of calls …" and "A1 L2/3 responses were contingent on the presence of specific spectrotemporal features". This is an interpretation, not a result. Either the analysis needs to be extended to support this statement in the Discussion (see comment 6; for example by showing a non-linearity in the RFs), or the authors should better clarify what is a result and what is interpretation.

Reviewer #2: In the study authors conducted experiments, in contrast to most previous studies, in unanesthetized animals, used an extensive set of conspecific calls as stimuli, compared

the responses to these stimuli across thalamic and cortical processing stages, and separated A1 neurons recorded from thalamorecipient and superficial layers. They found that whereas call representations in vMGB and A1 L4 were similar, a critical transformation occurs between A1 L4 and A1 L2/3. While vMGB and A1 L4 neurons seemed to respond primarily to the spectral content of calls resulting in a dense representation of calls, A1 L2/3 responses were contingent on the presence of specific spectrotemporal features, resulting in a highly sparse representation of calls.

This is a well performed study, showing experimental confirmation of some ideas that were present in the previous study of authors (Liu et al., 2019). It shows for the first time that in guinea pig (and probably in other animals as well) A1 exists essential neuronal interface between A1 L4 and A1 L2/3 that is responsible for detection of specific features of animal´s vocalization. 

I have following comments and questions to authors: 

i/ Grimsley et al. (2012) demonstrated in narcotized guinea pigs that "the primary area (AI) and three adjacent auditory belt areas contain many units that give isomorphic responses to vocalizations". According to them "area VRB (ventrolateral belt) has a denser representation of cells that are better at discriminating among calls by using either a rate code or a temporal code than any other area". Although the authors mention the paper by Grimsley et al. (2012) in their manuscript, I believe it deserves more attention in the Discussion. Speculations, how information about the specific features of animal´s vocalizations, as detected in the A1 L2/3 by the authors, will be transmitted and used in the secondary areas of the guinea pig´s auditory cortex will be appreciated. 

ii/ Traditionally different types of vocalizations are illustrated by their spectrograms and waveforms. This information is missing in the manuscript, although the results are based on 8 different vocallzation categories. The spectrograms and waveforms of vocalizations used by the authors could be shown in the Supplementary material. 

iii/ Fig. 3D shows joint distributions of the number and length of response windows. vMGB and A1 L4 neurons exhibit either multiple short windows or a single long window. In contrast, A1 L2/3 neurons exhibited one or two short response windows. The significant difference between vMGB and A1 L2/3 and between A1 L4 and A1 L2/3 is not surprising but why is there large difference between vMGB and A1 L4? This finding deserves at least some explanation in the Discussion.

iv/ There is some inconsistency in the presented illustrations and descriptions of the call selectivity/responsiveness. Fig. 1G suggests there are about the same portions of neurons responding to individual calls in all structures, i.e. the overall ratio of responsiveness between MGB and A1 L2/3 is about 1:1. In contrast to this Fig. 3A suggests that the responses to calls are less frequent in the A1 L2/3 than in MGB with the ratio about 4.5 : 6.5. It is not easy to understand how the values for Fig. 1G have been calculated (e.g. for 28 MGB neurons should be the blue bars of percentage with the step of 100/28 = 3.6%, but the differences among the blue columns are much lower). 

In addition, the Fig. 1G is also used as a proof that the illustrated effect of the emergence of call selectivity in A1 L2/3 is not due to only some of the call types (lines 295-300) " This was not the case in our data - neural preference for call type was evenly distributed across all tested call types across the processing stages (Fisher's exact test [37], p = 0.99; Fig. 1G)." 

This statement should be supported by some analysis (like a kind of bootstrap).

v/ It could be also interesting if responses to tone and/or noise stimuli are included into the analysis, to point out if the change in selectivity is specific for complex stimuli (like calls).

vi/ Line 610. How many neurons from individual structures responded to vocalizations by inhibition?

Minor points: 

Line 93. This sentence requires citation. 

Line 153. Please explain what means LFP responses.

Line 513. How about placement of the chamber for electrophysiological recording in the case of MGB?

Line 519. ….at a sampling rate of 100 kHz … 

Fig. 1D. Please explain what means CSD. 

Reviewer #3: Montes-Lourido et al investigate the emergence of stimulus category selectivity across the thalamocortical processing hierarchy for species-specific vocalizations in guinea pigs, and how this impacts the information encoded by neurons. Through experiments conducted in awake, head-fixed animals, the authors claim that neurons in the lemniscal region of the auditory thalamus (vMGB) and thalamocortical input layer of primary auditory cortex (A1 L4) are not selective to specific calls, but that superficial layers L2/3 of A1 "abruptly" shows much more selectivity to call features. That there would be some layer-dependent differences in processing stimuli is well-expected and even demonstrated in published works. Much less is understood though about these differences and their functional importance in the case of processing species-specific vocalizations. Thus, the topic is of interest to auditory neuroscientists, and of potential relevance for sensory cortical researchers interested in ethological stimuli more generally. The manuscript is laid out and written clearly, with generally rigorous analyses and transparent presentation of data. However, conceptual, methodological and interpretational concerns reduce my enthusiasm and leave me unconvinced at this point.

Major points:

1) The authors provide good transparency on much of their neural data, but there is critically missing information about the spectral content of the vocalizations used in their study and its overlap with the best frequencies of neurons recorded. This is problematic because of potential differences in their L2/3 vs. L4 and vMGB neural populations. Many gerbil vocalizations fall heavily in the 2-10 kHz range, and especially around 3-6 kHz. This is a frequency range where there is a lack of best frequencies in the recorded population of L2/3 neurons, unlike for L4 and vMGB (Fig 1F). Even though similar percentages of neurons across areas respond to the different calls (Fig 1G), if the same types of neurons are not being sampled, then either those BFs don't exist in L2/3 (potentially interesting but unlikely given results in Grimsely et al, 2012), or else the conclusion that there are L4 vs. L2/3 differences rests on an artifact of undersampling, and would not hold up. 

2) One place where this problem could lead to a misinterpretation is shown in Fig 3A. The main difference between A1 L2/3 and L4 appears to be the more uniform distribution across category number of the former vs. the more peaked distribution towards more categories of the latter. But if the BFs of L2/3 neurons are missing a spectrally dense region of the vocalizations, then it could make sense that there are not as many neurons whose responses track those acoustics as well as in the L4 or vMGB populations. Analyzing the response categories as a function of how neurons are tuned might be one way to get at this, but filling in the distribution of BFs around 3-6 kHz would be best. 

3) Related to above, the authors make the claim that L2/3 differences cannot be attributed to low-level differences in tuning properties. However, they only use tuning bandwidth as a comparison, and Fig 1F seems to show that L2/3 tuning in the 6-12 kHz BF range, where many of the calls probably have spectral content, looks to be particularly narrow compared to L4. A more thorough comparison of tuning properties as a function of best frequency would be helpful. One could also look at predicting the response to calls based on the tuning curve itself. Note though that even if it were the case that tuning per se is not explanatory of the responses for some neurons, that would be consistent with findings from Eli Nelken's lab many years ago looking at how some cat auditory cortex neurons respond unexpectedly to the spectral context around bird vocalizations. 

4) Conceptually, the authors push the idea that there is an "abrupt" transition between the way L4 and L2/3 of A1 encode these vocalizations. This is a potentially novel point. However, while the authors make a reasonable case that there are differences, whether those differences should constitute an "abrupt" change in coding or not is subjective. They do not provide any sense of what a "gradual" change would look like, and whether there is a quantitative way to differentiate "abrupt" from "gradual." In fact, many of the distributions comparing L4 to L2/3 look different, and while some figures show vMGB is not different from L4 (Fig 3A and B), others do not show vMGB data (Fig 4) at all, ignore differences between vMGB and L4 (Fig 3D), or do not indicate whether apparent differences are significant or not (Fig 3G). 

5) The authors claim, "A1 L2/3 responses were contingent on the presence of specific spectrotemporal features, resulting in a highly sparse representation of calls." However, besides showing a sparse but higher probability response across repeated trials at specific points in the stimulus, the authors do not really provide much to support this claim. It would make the paper stronger to show this, but at this point it is speculative. Presumably, many of these calls have rhythmically repeated elements. What are the spectrotemporal elements that the L2/3 neurons are responding to and why don't they respond when those same elements occur elsewhere in the sound?

6) The authors have described a creative way to automatically identify putative regions of the spiking responses. However, they should start first by comparing overall evoked firing rates in the different areas; that by itself may show the main difference between regions (or if not, then their case for focusing on kurtosis and sparseness is stronger). Furthermore, there is some arbitrariness to how they arrive at their bounding boxes and how regions get merged together. The authors should test whether their conclusions change as they change their criterion. For example, the neuron in Fig 2B (right) has many bounding boxes in response to a Chut, while the one in Fig 2B (left) looks like it has a similar pattern of responses to a Chirp. Yet in the former, the response is broken into many bounding boxes, while they are all merged together for the latter. This seems arbitrary based on an experimenter's empirical algorithm rather than what the brain may care about. For low firing neurons such as the one in the middle of Fig 2C, there are responses to Other or Rumble calls that are not merged together under the current criterion, but might have been if the criteria were more permissive to capture the rest of what appears to be an excitatory response. 

7) The authors report spontaneous activity differences in the Discussion, noting a p-value of only 0.03. This is a result, and should be put into the results section along with a plot of the distributions of the spontaneous activity across areas, just as for Fig 3 and 4, especially since they make an argument in the Discussion comparing spontaneous activity to other species. The concern is that the spontaneous activity is actually quite variable in all these areas, and that is not being reflected in the examples of Fig 2.

8) Methodologically, the authors calculate stimulus-specific information by filling in responses after the end of stimuli of varying lengths with simulated spontaneous firing. Why? This seems completely unnecessary if the goal is to understand the true nature of how neurons are conveying information about stimuli. If the actual firing after the end of a stimulus is spontaneous, then that should be present in the trial-by-trial firing already; if not, then the results from simulating firing do not reflect the reality. It is also unclear why time bins before the start of the stimulus are used. 

9) While the information analyses take into account suppressed portions of the response, the data being analyzed consists only of neurons whose firing has an excitatory component, based on having an evoked rate 6 standard errors from the mean spontaneous activity in one of its windows. Prior research in awake mice has shown the importance of neurons that are fully suppressed by vocalizations. Presumably such neurons exist in the guinea pig as well, and some estimate of how often they are encountered in L4 or L2/3 would be useful. The information theoretic analysis could also include such neurons to give a more complete view.

10) The Discussion is generally written in a scholarly manner with good coverage of existing literature in the guinea pig and nonhuman primate. However, the authors ignore relevant studies in other rodent species, including rats and mice. Moreover, several points are overstated.

- The authors claim an advance in using an unanesthetized preparation to study vocalization processing, but this has been done for many years in rats and mice. 

- The authors claim an advance in using an extensive set of conspecific calls, but again there are many papers that do so in rats, mice and gerbils. 

- The authors reference work from the Portfors lab to argue that mice are an outlier in showing an over-representation of social vocalizations in subcortical stations in a way that is not consistent with pure tone tuning. That argument is overstated, especially given other work from the Lesica lab using dense recording methods showing that responses to calls are largely consistent with best frequencies of neurons, as with other species. Furthermore, other species besides mice also show "associative and behavioral functions" in the responses of auditory cortical neurons. Hence, aside from the fact that guinea pig vocalizations are lower frequency than mouse ultrasonic calls, there is no reason to claim that mice are an "outlier."

11) The authors make the bold claim that their results indicate that "critical transformations to sound representations occur at the A1 L4 → A1 L2/3 synapse." There is simply no evidence to support that. All the studies reported here are based on extracellular recordings broken down by depth, with layer inferred from a current source density analysis. There are no whole cell recordings, cross-correlation analyses of simultaneously recorded units, etc. that would be expected to claim anything about the specific quoted synapse. In fact, based on studies in other rodents (e.g. Barbour and Calloway, Journal of Neuroscience, 2008), L2/3 neurons receive input not only from L4, but also from L2/3 and L5. What gives rise to the different coding of L2/3 vs. L4 neurons thus need not happen within just one synapse, especially given that the sparse response windows often occur late in the stimulus. The author's intuition may well be correct, but here they provide no evidence in support of that case.

Other points

- Fig 1F - the experiments are based on recordings from 5 animals. To give more confidence that the results are not dominated by just a few animals, the best frequency plots could be modified to show different symbols for different animals so that the reader can weigh how much any one animal contributes to the results. 

- Because the L2/3 distribution of call-selectivity is more uniform across the number of call categories rather than being peaked at 1 or 2, there are just as many neurons in L2/3 that are NOT call-selective. The language throughout should be toned down so that this is clearer, rather than giving the impression that one only finds call-selective responses in L2/3. 

- The authors state, "Surprisingly and contrary to our expectation, we found even lower MI levels over the population of A1 L2/3 neurons (Fig. 4A, red)." I am not clear why this should be considered surprising. It fits with the lower firing rate and sparseness of the activity in L2/3. 

- Fig 3E should show an example vMGB neuron as well

- Fig 4 should show analyses for vMGB as well

- "Data shows" is incorrect - "data" is plural

Finally, concerning the use of statistics, the K-S test is fine for making claims that distributions are different, but it can be overly sensitive and lead to false positives. The use of the FDR correction is a not-particularly-conservative way to account for that, but still does not provide a way to claim anything about whether there is an "abrupt" difference or not between distributions.

---

## [Decision Letter · Decision Letter 2]

11 May 2021

Dear Dr Sadagopan,

Thank you very much for submitting a revised version of your manuscript "A complex feature-based representation of vocalizations emerges in the superficial layers of primary auditory cortex" for consideration as a Research Article at PLOS Biology. This revised version of your manuscript has been evaluated by the PLOS Biology editors, the Academic Editor and the original reviewers.

The reviews are appended below. As you will see, both reviewers 1 and 2 think the revision has addressed their previous comments and they think the manuscript is substantially improved. However reviewer 3 still has a number of lingering concerns which will need to be addressed before we can consider the study for publication.

In light of the reviews, we are pleased to offer you the opportunity to address the remaining points from the reviewers in a revised version that we anticipate should not take you very long. We will then assess your revised manuscript and your response to the reviewers' comments and we may consult the reviewers again.

Along with addressing the remaining reviewer comments, we also ask that you address the following editorial requests:

1) Ethics request: In your methods section, please include the identification number of your protocol, approved by the University of Pittsburgh IACUC.

2) Data request: Please provide, as a supplementary file, the underlying data for each figure in your study. Please also reference this dataset in the figure legends. For example, to each figure legend you might add the following statement: “data underlying this figure can be found in supplementary file S1_data.” You will also need to ensure that this data file contains a legend, and is referenced in your data availability statement. I have included more information regarding our data sharing policy and this request below my signature. 

3) Thank you for changing your title in response to reviewer comments in the last round of review. We have been discussing the title of your manuscript, and wonder if it might be edited slightly to be more accessible to a broader readership? If you agree, you might change it to something like 'Neuronal selectivity to specific vocalizations emerges in the superficial layers of primary auditory cortex'

4) Please take a moment to review your reference list to ensure that it is complete and correct. If you have cited papers that have been retracted, please include the rationale for doing so in the manuscript text, or remove these references and replace them with relevant current references. Any changes to the reference list should be mentioned in the cover letter that accompanies your revised manuscript. 

We expect to receive your revised manuscript within 1 month.

**IMPORTANT - SUBMITTING YOUR REVISION**

*Resubmission Checklist*

*Published Peer Review*

*PLOS Data Policy*

*Blot and Gel Data Policy*

Sincerely,

Lucas Smith

Associate Editor

PLOS Biology

lsmith@plos.org

DATA POLICY REQUEST:

Figure 1 B,D-H; Figure 2; Figure 3A-C; Figure 4A-E; Figure 5C-I; Figure 6A,C,E-G,I,K; Figure 7A-D; Figure 8A-K; Figure S1A-D;

**IMPORTANT: Please also ensure that figure legends in your manuscript include information on where the underlying data can be found, and ensure your supplemental data file/s has a legend.

**IMPORTANT: Please ensure that your Data Statement in the submission system accurately describes where your data can be found.

REVIEWS:

Reviewer #1: 

I would like to thank the authors for conducting extensive additional analyses, which confirm the conclusions they drew in the original version of this manuscript. I have no further comments.

Typo in lines 174, 176, 413: ANOCOVA  ANCOVA

Reviewer #2: All my comments on the previous version were taken into account, the manuscript was substantially improved and I do not have other comments. 

Reviewer #3: The manuscript from Montes-Lourido et al is improved and makes a more convincing case for a transformation in the representation of species-specific vocalizations between cortical layers in the guinea pig. The authors are creative in their analyses of the responses and have generally provided transparent and rigorous statistical analyses to illustrate their points. Their new Table nicely summarizes differences between the areas, though statistically speaking, it would make more sense to compute effect sizes and list those rather than the degree of significance in a statistical test. 

However, one issue that remains a serious concern is the mutual information (MI) analysis, which has been expanded to include data from vMGB in addition to A1 L4 and L2/3. The authors' explanation of having to "fill-in" the spontaneous activity to compute the MI is appreciated, but their full explanation did not make it into the Methods, and that should be updated. It would also be helpful to the reader to know exactly which stimuli received these filled-in spikes (Purr?) since there are examples in Fig. 8F-H that seem to show abrupt drops in SSI about 1 second after the end of the stimulus. Given recent studies of auditory cortical Off responses after the end of stimuli, the act of substituting spiking could potentially skew results. Checking whether neurons exhibit firing rate differences over the last 1 sec vs. the simulated 1 sec of spontaneous activity, and how that affects the calculated MI, would help assuage this concern, but at the least some clearer statement about what was done and its justification in the Methods seems warranted.

More seriously, although the filling-in explanation now helps me understand what the authors did in computing the MI, I am left somewhat perplexed by how they then use these results to support their conclusions. The authors make a point about longer temporal integration times in L2/3 based on the area-under-the-curve of their bin-by-bin mutual information curves. This seems problematic. The Methods indicate that they use a sliding window with overlap between points, meaning that each time bin is not independent from each other. Moreover, the MI can obviously be negative because of their shuffle correction subtraction (e.g. Fig 8C), but it is not clear what constitutes the AUC in that case. In general, there is no obvious physical or conceptual meaning to the AUC of their MI curve aside from it being used to show a difference between neural regions. If the authors intended the AUC to essentially represent the average MI over the duration of the sounds, then at the least they should eliminate the non-independent points that overlap in that average. Even still though, computing an average MI for a neuron in this fashion is questionable because of the nonstationarity of the stimulus over the time bins and correlations in firing between time bins. A methodologically more sound way to estimate MI would be to divide time windows into smaller bins and create spike words. It takes many trials to estimate the probability distribution across words though, and it is not clear whether enough trials were run to do this well. Estimating MI based on spike counts (not words) in large windows as done by the authors may be ok (though does not get at "activity patterns" as they claim), but it is still not clear what averaging or summing the values from adjacent windows (as was done to compute the AUC) means. 

As far as supporting a point about integration time, Fig. 8B does show that L2/3 looks different as a function of the window length, but there are no error bars, and no explanation of what normalization was used. Thus, it is hard to know what exactly to make of the shift to the right of that curve and why the three curves meet at 400 ms. The authors seem to be arguing that because the AUC for L2/3 neurons does not saturate above 200 ms, the integration time for L2/3 is longer than in the other two brain areas. This argument is based on the rate of change of the AUC with increasing window length, and exactly how that relates to the integration time is unclear. If anything, because the authors are essentially counting spikes in ever larger windows for their MI calculation, the exact timing of the spikes within the window becomes less important as the window size increases, so the L2/3 result seemingly says more about worsening temporal precision rather than how long a stimulus window is needed to drive responses (integration time). However, that interpretation seems to be at odds with the conclusions from Fig 4B, where the authors argue that the feature response window for L2/3 neurons is actually smaller than in L4 and vMGB. Hence, the concern is that the AUC approach to characterizing the MI and how temporally extended "patterns" of firing conveys information is leading to difficult to interpret results, and the authors should reconsider whether to include it.

Finally, also in the context of computing information, the authors refer to high-MI points, and highlight these in Fig 8C-E with red crosses. However, the Methods do not explain how these were chosen; they appear to be the peaks, but not all peaks are selected. Since high-MI points were used in their subsequent correlation analysis with the PSTH, it is important to explain this in detail. Are their results changed if they use all bins rather than just those with high MI?

---

## [Editor Report · Decision Letter 3]

24 May 2021

Dear Dr Sadagopan,

On behalf of my colleagues and the Academic Editor, Manuel Malmierca, I am pleased to say that we can in principle offer to publish your Research Article "Neuronal selectivity to complex vocalization features emerges in the superficial layers of primary auditory cortex" in PLOS Biology, provided you address any remaining formatting and reporting issues. These will be detailed in an email that will follow this letter and that you will usually receive within 2-3 business days, during which time no action is required from you. Please note that we will not be able to formally accept your manuscript and schedule it for publication until you have made the required changes.

Together, with the Academic Editor, we have discussed your revision and think that you have satisfactorily addressed the reviewer concerns, and the majority of our editorial requests. As one last, minor editorial request, we ask that you provide a legend for each of supplementary data files that you generated, containing the data underlying your figures. An example legend might be: "S1_Data. Data underlying figures ___." You can add these legends while addressing the formatting and reporting requests which you will receive from our productions team.

PRESS

Thank you again for supporting Open Access publishing. We look forward to publishing your paper in PLOS Biology. 

Sincerely, 

Lucas Smith, Ph.D. 

Associate Editor 

PLOS Biology